# From plasmodesma geometry to effective symplasmic permeability through biophysical modelling

Eva E Deinum[1]*, Bela M Mulder[2,3]†, Yoselin Benitez-Alfonso[4]†

[1]Mathematical and statistical methods (Biometris), Wageningen University, Wageningen, Netherlands; [2]Living Matter Department, Institute AMOLF, Amsterdam, Netherlands; [3]Laboratory of Cell Biology, Wageningen University, Wageningen, Netherlands; [4]Centre for Plant Science, University of Leeds, Leeds, United Kingdom

**Abstract** Regulation of molecular transport via intercellular channels called plasmodesmata (PDs) is important for both coordinating developmental and environmental responses among neighbouring cells, and isolating (groups of) cells to execute distinct programs. Cell-to-cell mobility of fluorescent molecules and PD dimensions (measured from electron micrographs) are both used as methods to predict PD transport capacity (i.e., effective symplasmic permeability), but often yield very different values. Here, we build a theoretical bridge between both experimental approaches by calculating the effective symplasmic permeability from a geometrical description of individual PDs and considering the flow towards them. We find that a dilated central region has the strongest impact in thick cell walls and that clustering of PDs into pit fields strongly reduces predicted permeabilities. Moreover, our open source multi-level model allows to predict PD dimensions matching measured permeabilities and add a functional interpretation to structural differences observed between PDs in different cell walls.

*For correspondence:
eva.deinum@wur.nl

†These authors contributed equally to this work

Competing interests: The authors declare that no competing interests exist.

## Introduction

The formation of spatial patterns in plants requires the transport and interaction of molecular signals. This sharing of information coordinates cell fate decisions over multiple cells and the isolation of cell fate determinants within a cell or group of cells on the same developmental path. Small molecules such as sugars, peptides, hormones and RNAs move long and short distances to coordinate cell/organ development (*Otero et al., 2016*). Cell-to-cell transport of proteins, such as transcription factors, is also important in the regulation and/or developmental reprogramming of local cellular domains (*Gallagher et al., 2014*). A well studied example is SHORT-ROOT (SHR), an *Arabidopsis thaliana* GRAS family transcription factor, that moves from the stele to cortical-endodermal tissue layers to specify cell fate and root patterning (*Nakajima et al., 2001*; *Spiegelman et al., 2018*; *Wu and Gallagher, 2013*; *Wu and Gallagher, 2014*). Other mobile factors with developmental importance include TARGET OF MONOPTEROS 7, PEAR transcription factors and miRNAs (*Lu et al., 2018*; *Miyashima et al., 2019*; *Skopelitis et al., 2018*).

Plant cells are connected by channels named plasmodesmata (PDs) that facilitate the transport of these molecules. PD are narrow membrane lined structures embedded in cell walls to allow for symplasmic (cytoplasm-to-cytoplasm) molecular flux (*Figure 1*). The ER forms a tubular structure called desmotubule (DT) that traverses the PD, leaving a discrete cytosolic sleeve (also called 'cytoplasmic sleeve' in the literature) where molecular transport occurs (*Nicolas et al., 2017a*; *Sager and Lee, 2018*). In the region closest to the PD entrances, the cytosolic sleeve appears constricted (neck) in most tissue types, although there are recent observations of 'straight' PDs in meristematic root

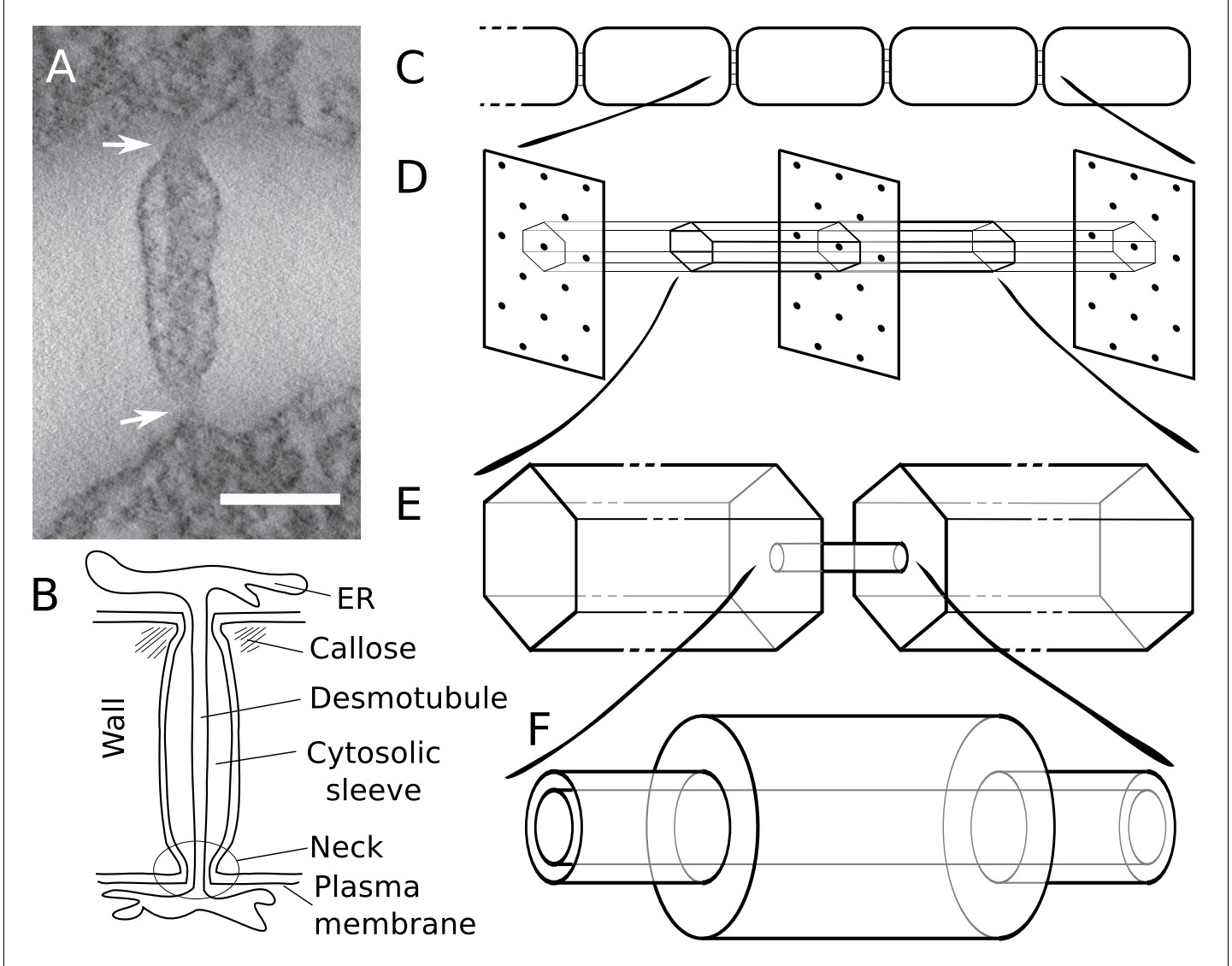

**Figure 1.** Modelling effective symplasmic permeability: concept overview. (A) Electron microscopy image showing one PD, constricted at the neck regions (arrows), from *Arabidopsis thaliana* root tissue. The image was extracted from a reconstructed tomograph. Scale bar: 50 nm. The image was kindly provided by the Bayer lab. (B) Cartoon showing PD geometry and structural features. (C-F) The model to determine effective symplasmic permeability considers that connectivity within a cell file (C) is affected by the distribution of PDs in the cell wall (D) (modelled as a function of the cytoplasmic column belonging to a single PD (E)) as well as by the structural features of individual PDs (F).

sections (*Nicolas et al., 2017b*). Cell walls at PD locations play a key role in regulating its dimensions. The accumulation of callose, a cell wall beta-1,3 glucan polysaccharide synthesized by callose synthases and degraded by β−1,3-glucanases (*Zavaliev et al., 2011*; *Amsbury et al., 2017*), is the best understood mechanism for the control of PD dimensions and symplasmic transport capacity (i.e. effective symplasmic permeability). Other factors such as membrane composition, shape and number of PDs change during development and between cell types adding extra dimensions to PD regulation (*Nicolas et al., 2017a*). Mutants blocked in PD form and function are embryo or seedling lethal, highlighting the importance of these structures for normal plant development (*Kim et al., 2002*; *Benitez-Alfonso et al., 2009*; *Xu et al., 2012*).

Small molecules can move via PD by diffusion (non-targeted transport). This is considered to be predominantly symmetrical (*Schönknecht et al., 2008*; *Maule, 2008*), while in certain tissues, such as secreting trichomes (*Waigmann and Zambryski, 1995*; *Gunning and Hughes, 1976*) and the phloem (*Ross-Elliott et al., 2017*; *Comtet et al., 2017*), hydrodynamic flow may create

directionality. The maximum size of molecules that can move by this generic 'passive' pathway is often referred to as the 'size exclusion limit' (SEL), which obviously depends on PD properties and structural features (*Dashevskaya et al., 2008*). Large molecules can move through PD via an 'active' or 'targeted' pathway overriding the defined SEL. This may involve additional factors that temporarily modify these substrates, target them to the PDs, or induce transient modifications of the PDs to allow for the passage of larger molecules in a highly substrate dependent fashion (*Zambryski and Crawford, 2000*; *Maule et al., 2011*).

Computational modelling approaches have been applied to model PD transport but, so far, these have mainly focused on hydrodynamic flow and the specific tissues where that matters (*Blake, 1978*; *Bret-Harte and Silk, 1994*; *Jensen et al., 2012*; *Ross-Elliott et al., 2017*; *Comtet et al., 2017*; *Foster and Miklavcic, 2017*; *Couvreur et al., 2018*). The few existing studies on diffusive transport do not consider neck constrictions or the approach to PDs from the cytoplasmic bulk. Most models consider PDs as straight channels, with advective/diffusive transport through an unobstructed cytosolic sleeve and typically, but not always, account for reduced diffusivity inside these narrow channels compared to the cytosol (*Bret-Harte and Silk, 1994*; *Liesche and Schulz, 2013*; *Dölger et al., 2014*; *Ross-Elliott et al., 2017*; *Couvreur et al., 2018*). Only the oldest of this set, (*Blake, 1978*), uses a dilated central region in its calculations, but is entirely focused on hydrodynamics. In specific contexts, also a few other geometries are considered. (*Ross-Elliott et al., 2017*) also consider 'funnel' shaped PDs, which are observed in the phloem unloading zone, but ignore the DT in their diffusion model, as they only calculate a best case scenario for diffusive transport. In the context of size selectivity for small (sugar) molecules in phloem loading, also the so-called 'sub-nano channel model' of PD geometry has been considered (*Liesche and Schulz, 2013*; *Comtet et al., 2017*). In this model, symplasmic transport is modelled to be confined to nine cylindrical channels spanning the PD. This was based on a 9-fold rotational symmetry in enhanced 'top view' electron micrographs but never validated experimentally in longitudinal sections. Instead, sparsely spaced axial spoke structures have been reported (*Ding et al., 1992*; *Nicolas et al., 2017b*).

Experimental measurement of the parameters that determine effective symplasmic permeability is difficult and many examples exist of misleading and/or conflicting results. Generally speaking two main approaches are used, providing results at different scales that are hard to reconcile. On the one hand, ultrastructural observations using transmission electron microscopy (EM) can provide useful data on PD dimensions and structural features but, despite recent advances, sample preparation affects the integrity and dimensions of PDs to an unknown extent potentially yielding an underestimation of relevant parameters (*Nicolas et al., 2017b*). On the other hand, tissue level measurement of symplasmic fluxes is achieved using symplasmic molecular reporters, but this is either invasive or limited to few molecular sizes, mostly fluorescein and its chemical relatives (hydrodynamic radii of about 0.4 to 0.6 nm) and GFP derived fluorescent proteins (such as DRONPA-s (28 kDa), Dendra2 (26 kDa), (photoactive and non-photoactive) single GFP (27 kDa, hydrodynamic radius 2.45–2.82 nm) and its multiples [*Calvert et al., 2007*; *Terry et al., 1995*; *Chudakov et al., 2007*; *Gerlitz et al., 2018*; *Kim et al., 2005*; *Rutschow et al., 2011*]). In all cases, the tissue geometry and varying degrees of vacuoloarization can severely complicate the interpretation of the measurements in terms of effective wall permeability for symplasmic transport. Old data on symplasmic permeability use either microinjection or particle bombardment, which allow for a much wider size range of dyes/molecular reporters, but these techniques can produce cellular stress, which affects PD function (*Liesche and Schulz, 2012*). Even when using the same dye/fluorescent molecule and the same tissue, these approaches deliver much lower permeabilities than less invasive techniques, demonstrating that they are unreliable for estimating permeabilities in unperturbed plants (e.g. see *Haywood et al., 2002*, or compare *Rutschow et al., 2011* and *Goodwin et al., 1990*). Less invasive methods involve transgenic lines expressing fluorescent proteins under cell-specific promoters (*Roberts et al., 2001*; *Stadler et al., 2005a*), which are very time consuming to generate, or photo-activation and photobleaching techniques (*Rutschow et al., 2011*; *Gerlitz et al., 2018*). These approaches have yielded valuable insights, but again, both are limited to few proteins/molecular sizes.

In summary, despite recent advances in the development of probes and techniques, effective symplasmic permeability is difficult to assess directly. The fast response of plants to wounding and other stresses, may render part of the ultrastructurally derived parameters less reliable than others, explaining the frequent observation of apparently incompatible results when modelling diffusive

symplasmic transport from ultrastructural measurements. In a multi-species analysis correlating photobleaching and electron microscopy results, (*Liesche et al., 2019*) were unable to find a universal model for matching measurements at the ultrastructural and tissue levels for different interfaces along the phloem loading pathway, illustrating the need for better models. Ideally, we would be able to integrate the results of the experimental approaches at both levels in a model that considers their limitations in order to get more accurate estimates of effective symplasmic permeability and the underlying structural parameters. This brings us to our central question: what do we need to assume about PD size, number, structure, etc. to be able to reproduce tissue level measurements? Moreover, PD geometry changes during development (*Roberts et al., 2001*; *Fitzgibbon et al., 2013*), inspiring our second main question: how do distinct features of PD geometry influence transport properties?

Here, we describe the biophysical properties of diffusive symplasmic transport considering detailed PD structural features (such as the DT and the neck region) and the approach from the cytoplasmic molecular bulk towards PDs that are either evenly distributed or clustered into pit fields (*Faulkner et al., 2008*) (*Figure 1*). Inside our model PDs, the entire cytosolic sleeve is available for particle diffusion ('unobstructed cytosolic sleeve model'). We investigate how neck/central region, wall thickness, the presence of a DT and PD clustering into pit fields affect transport characteristics for different particle sizes, adding a functional context to some puzzling recent experimental observations. We also apply our framework to compute effective permeabilities for carboxyfluorescein (CF), a fluorescent dye used routinely to measure changes in symplasmic permeability. Comparing calculated and experimentally measured values, we demonstrate that the relatively high effective CF permeabilities observed by *Rutschow et al. (2011)* can be explained by our model of diffusive non-targeted symplasmic transport and reveal the potential source of conflicts with ultrastructural measurements. We found that, in this context, our model performed better than the 'sub-nano channel model' (*Liesche and Schulz, 2013*) referred to above. Our calculations demonstrate that multi-scale modelling approaches can integrate results from PD structural dimensions and molecular fluxes and reveal conflicts on these determinations. We, therefore, recommend these should be applied systematically when defining effective symplasmic permeability for a particular tissue/molecule and/or biological context. To facilitate this, we share a python program for computing effective permeabilities from PD geometries as a community resource.

## Results

### Outline of the model

Our aim is to describe the symplasmic transport properties of a cell wall as an effective wall permeability, that is a single number that could be plugged into tissue/organ level models. For this, we split the transport into two parts: the movement through an individual channel representing a PD and the approach to this channel from the cytoplasmic bulk (*Figure 1*). This implicitly assumes a homogeneous cytosol. The basic geometrical terminology that we considered in our calculations is introduced in the cartoon PD shown in *Figure 1B*. An overview of all mathematical symbols is given in Appendix 1.

Obtaining good EM data of PD dimensions is notoriously hard. We therefore opted for a simple geometrical description that allows us to study the effects of PD neck, central region and desmotubule dimensions with as few parameters as possible (see Materials and methods). We modelled a single PD as a 3-part cylindrical channel (*Figure 2A*), with total length $l$, which would typically equal the local wall thickness. The ends of the channel were modelled by narrow cylinders representing the plasmodesmal 'neck' constriction. These have length $l_n$ and radius $R_n$. The central region has radius $R_c$. Over the whole length, the center of the channel is occupied by a 'desmotubule' (DT) modelled as a cylinder of radius $R_{dt}$. The part available for diffusive transport, the cytosolic sleeve, is the space between the outer cylinder wall and the DT.

We made the arguably simplest choice of modelling particles as (non-additive, i.e. not interacting among themselves) hard spheres with radius $\alpha$. This is partially supported by previous research showing that the hydrodynamic radius is the main determinant of PD transport characteristics, leaving behind, among others, particle charge (*Dashevskaya et al., 2008*; *Terry and Robards, 1987*). We also assumed that PD walls are rigid, and hence are unable to deform to accommodate larger

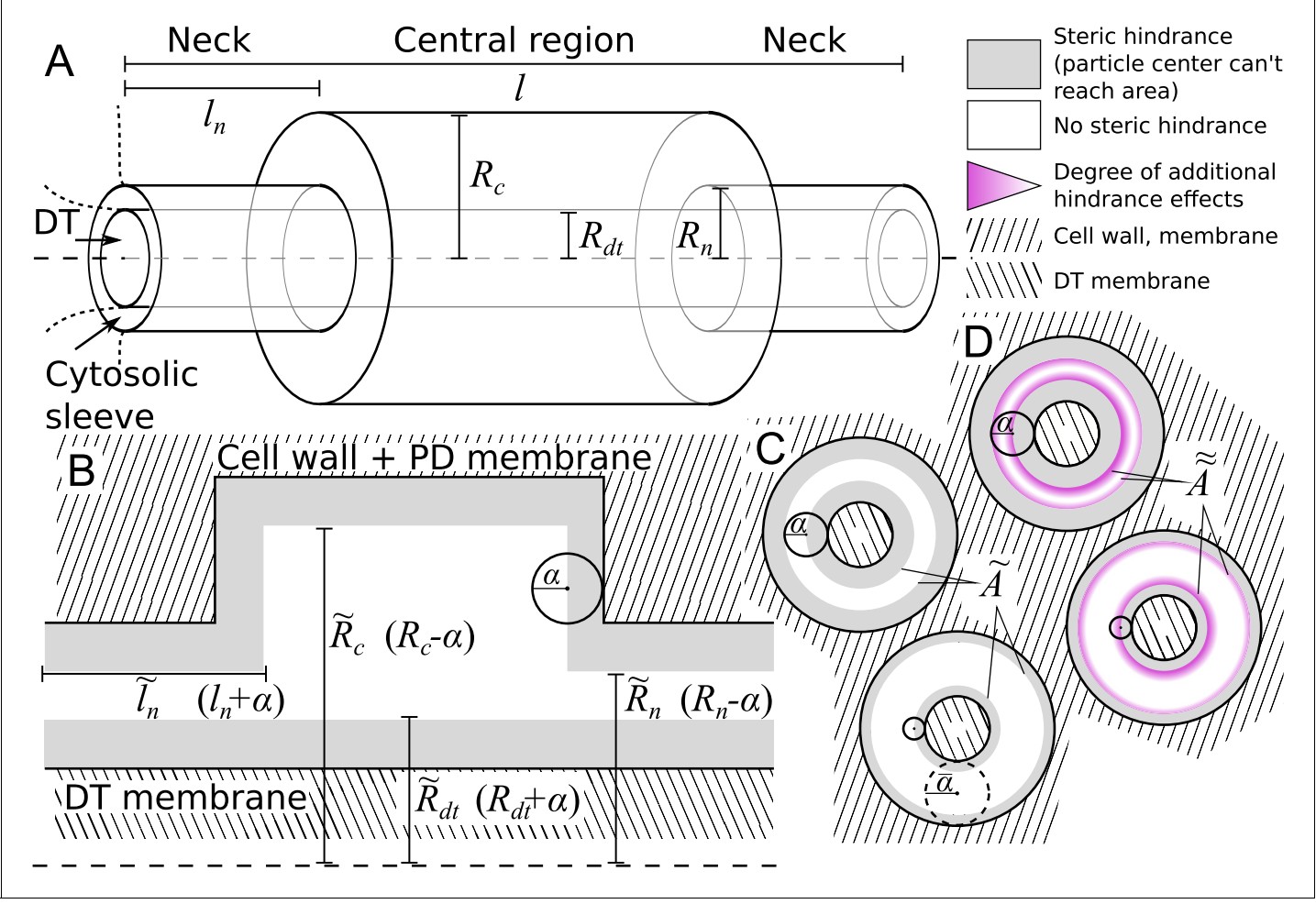

**Figure 2.** Model PD geometry and hindrance effects. (A) Individual PDs are modelled using multiple cylinders with a total length $l$, neck (inner) radius $R_n$ and neck length $l_n$, central region (inner) radius $R_c$ and desmotubule (outer) radius $R_{dt}$. B,C: Illustration of the impact of steric hindrance and rescaled parameters. The gray areas of the longitudinal (B) and transverse (C) sections cannot be reached by the center of the particle with radius $\alpha$ (steric hindrance). For a concise description of the available volume and cross section area, we use the rescaled lengths $\tilde{l}_n = l_n + \alpha$, $\tilde{R}_c = R_c - \alpha$, $\tilde{R}_{dt} = R_{dt} + \alpha$ and $\tilde{R}_n = R_n - \alpha$. (C) The cross section area available for diffusion on a transverse section was named $\tilde{A}$, which depends on the particle radius ($\alpha$). $\tilde{A}$ is the area of the white ring in each cross section. The maximum particle size $\bar{\alpha}$ is illustrated with a dashed circle. For a particle of size $\alpha = \bar{\alpha}$, $\tilde{A} = 0$. (D) In practice, particles spend less time diffusing close to the wall than farther away from it (hydrodynamic hindrance). Consequently, the area close to wall contributes less to diffusive transport, as illustrated with purple gradients. These additional hindrance effects are accounted for in $\tilde{\tilde{A}}$.

particles. These assumptions imply a boundary condition: the center of a particle cannot come closer to the wall than the particle's radius $\alpha$ (*Figure 2B,C*). This so-called steric hindrance reduces the volume that is available for diffusion of the particle's center in a size dependent manner. Moreover, the maximum particle radius that can pass the PD, $\bar{\alpha}$, is always well defined. In practice, a precise definition of the SEL in terms of molecule size/shape is hard to give, however, we can use $\bar{\alpha}$ to operationalize the SEL concept in a straightforward manner. To avoid confusion, however, we will consistently write $\bar{\alpha}$ when referring to our model.

We introduced rescaled geometrical parameters to account for the reduced available volume in a compact way: $\tilde{l}_n = l_n + \alpha$, $\tilde{R}_c = R_c - \alpha$, $\tilde{R}_{dt} = R_{dt} + \alpha$ and $\tilde{R}_n = R_n - \alpha$. With these, the available surface area (*Figure 2C*) is

$$\tilde{A}_x(\alpha) = \pi(\tilde{R}_x^2 - \tilde{R}_{dt}^2), \qquad (2\alpha < R_x - R_{dt}), \tag{1}$$

with $x = n$ for the neck and $x = c$ for the central region. In the typical situation that the neck is the narrowest part of the channel, the maximum particle radius that can pass is: $\bar{\alpha} = (R_n - R_{dt})/2$.

Considering pure diffusion without particle turnover inside the PD, particle flux through the channel is described by $\frac{\partial C_{xyz}}{\partial t} = D\nabla^2 C_{xyz}$, or in steady state: $D\nabla^2 C_{xyz} = 0$, with $C_{xyz}$ the position dependent particle concentration and $D$ the particle's diffusion constant inside the PD. Note that $D$ strongly depends on particle size. Assuming a homogeneous distribution of particle flux over (the available part of) each channel cross section, we can treat diffusion through the channel as a simple 1D problem along the channel axis (for the impact of this assumption, see Appendix 2). Particle mass conservation, as dictated by the steady state diffusion equation, then gives that the local concentration gradient at position $x$, $\nabla C_x$, is inversely proportional to the available surface area $A_x$, so $\nabla C_c = \tilde{A}_n/\tilde{A}_c \nabla C_n$. The total concentration difference over the PD, $\Delta C = C_l - C_0$ is accordingly distributed over the channel: $\Delta C = 2\tilde{l}_n \nabla C_n + (l - 2\tilde{l}_n)\nabla C_c$. The steady state molar flow rate $Q(\alpha)$ through each channel is proportional to the entrance cross section: $Q(\alpha) = -D\tilde{A}_n \nabla C_n$. Solving these equations for $\nabla C_n$ leads to:

$$Q(\alpha) = -\frac{D\tilde{A}_n\tilde{A}_c}{2\tilde{l}_n\tilde{A}_c + (l - 2\tilde{l}_n)\tilde{A}_n}\Delta C. \tag{2}$$

This result can be improved further by incorporating hydrodynamic interactions between particles and walls (*Figure 2D*). To that end we followed (*Liesche and Schulz, 2013*) in employing the so-called hindrance factors $0 \leq H(\lambda) < 1$, which are based on proper cross sectional averaging of particle positions over time, as described by *Dechadilok and Deen (2006)*. Based on geometrical considerations, we used the factors for a slit-pore geometry (see Materials and methods). These factors depend on the relative particle size $\lambda$. In our case, $\lambda = 2\alpha/(R_x - R_{dt})$. In the neck region, $\lambda = \alpha/\bar{\alpha}$. For the full expression and behaviour of $H(\lambda)$, see Materials and methods.

As $H(\lambda)$ already includes the effect of steric hindrance between wall and particle, we can adjust *Equation 2* by replacing every instance of $\tilde{A}_x$ with

$$\tilde{\tilde{A}}_x = H\left(\frac{2\alpha}{R_x - R_{dt}}\right)A_x. \tag{3}$$

For completeness, we note that the simplification of a uniform particle flux along the channel axis is violated near the neck-central region transitions, resulting in an error of a few percent (see Materials and methods for further discussion). We now define the permeation constant of a single PD, $\Pi(\alpha)$, through the rule rule steady-state flow rate = permeation constant × concentration difference, yielding

$$\Pi(\alpha) \equiv \frac{Q(\alpha)}{\Delta C} = \frac{D\tilde{\tilde{A}}_n\tilde{\tilde{A}}_c}{2\tilde{l}_n\tilde{\tilde{A}}_c + (l - 2\tilde{l}_n)\tilde{\tilde{A}}_n}. \tag{4}$$

We also defined $\tau$ as the corresponding estimate for the mean residence time (MRT) in the channel. Using a steady state mass balance argument this can be calculated as the number of particles in the channel divided by the number leaving (or entering) per unit of time (see Materials and methods for further description).

$$\tau(\alpha) = \int_0^l C_x \tilde{\tilde{A}}_x \mathrm{d}x / Q(\alpha) \tag{5}$$

Having defined the permeation constant of a single channel, the effective symplasmic permeability of the wall as a whole ($P(\alpha)$, the quantity that can be estimated using tissue level measurements) follows from the definition $J = P\Delta C$ ($\mathrm{steady state flux = permeability \times density jump}$):

$$P(\alpha) = f_{ih}\rho\Pi(\alpha), \tag{6}$$

with $\rho$, the density of PDs per unit wall area (number/ $\mu m^2$) and $f_{ih}$, a (density dependent) correction factor for the inhomogeneity of the wall ($0 < f_{ih} < 1$). The latter takes into account that the wall is, in fact, only permeable at discrete spots. To calculate $f_{ih}$, we considered a linear chain of cells of length $L$ that are symplasmically connected over their transverse walls (*Figure 1C*) and computed mean first passage times (MFPT) through a straight PD and a column of cytoplasm surrounding the PD. The column was determined by assigning every bit of cytoplasm to the PD closest to it. For a regular

triangular PD distribution, this results in a hexagonal column from the middle of one cell to the middle of the next, with a PD in its centre (*Figure 1D*). We then converted the MFPT to an effective wall permeability and compared the result with the uncorrected effective permeability computed as $\rho\Pi(\alpha)$ (as described in the Materials and methods).

As expected, $P(\alpha)$ depends on particle size. Two factors underlie this size dependence, which both affect $\Pi(\alpha)$: hindrance effects, which reduce the space available for particle diffusion, and the fact that the diffusion constant is inversely proportional to particle size: $D = d_1/\alpha$. *Figure 3A* and (*Figure 3—figure supplement 1*) show that hindrance effects have the strongest impact for particle sizes close to the maximum $\bar{\alpha}$, whereas the particle diffusion constant always has a large impact *Figure 3B*. For example, at $R_n = R_c$, the 50+ fold difference between $\alpha = 0.1$ nm and $\alpha = 2$ nm is reduced to a 3-fold difference when ignoring the particle size dependence of the diffusion constant.

Using the model presented here, we computed the effects of different PD structural features and changes in PD density and distribution on effective symplasmic permeability and its dependence on particle size as described below.

## A dilated central region increases molecular flux in thicker cell walls

Electron microscopy suggests that PDs often have a neck region of reduced radius in comparison to the central region. We investigated how a constricted neck region, or, similarly, a dilated central region, affects PD transport. For this, we compared transport properties while conserving the size selectivity (constant $\bar{\alpha}$). We investigated how both the transport volume (using *Equation 2*) and transport time ($\tau$ as above) change when the central region is dilated. To compare channels with neck and dilated central region (12 nm $= R_n \leq R_c$) with narrow straight channels ($R_n = R_c = 12$ nm), we define a relative molar flow rate as $Q_{rel} = Q_{dilated}/Q_{narrow}$ and similarly relative $\tau_{rel}$ (*Figure 4*). For a more detailed discussion of $\tau_{rel}$ and its computation, see Materials and methods and Appendix 2.

We then investigated how both $Q_{rel}$ and $\tau_{rel}$ change with increasing central region radius $R_c$ and how this depends on particle radius $\alpha$ and PD length $l$ (*Figure 4*). The panels A and B show that

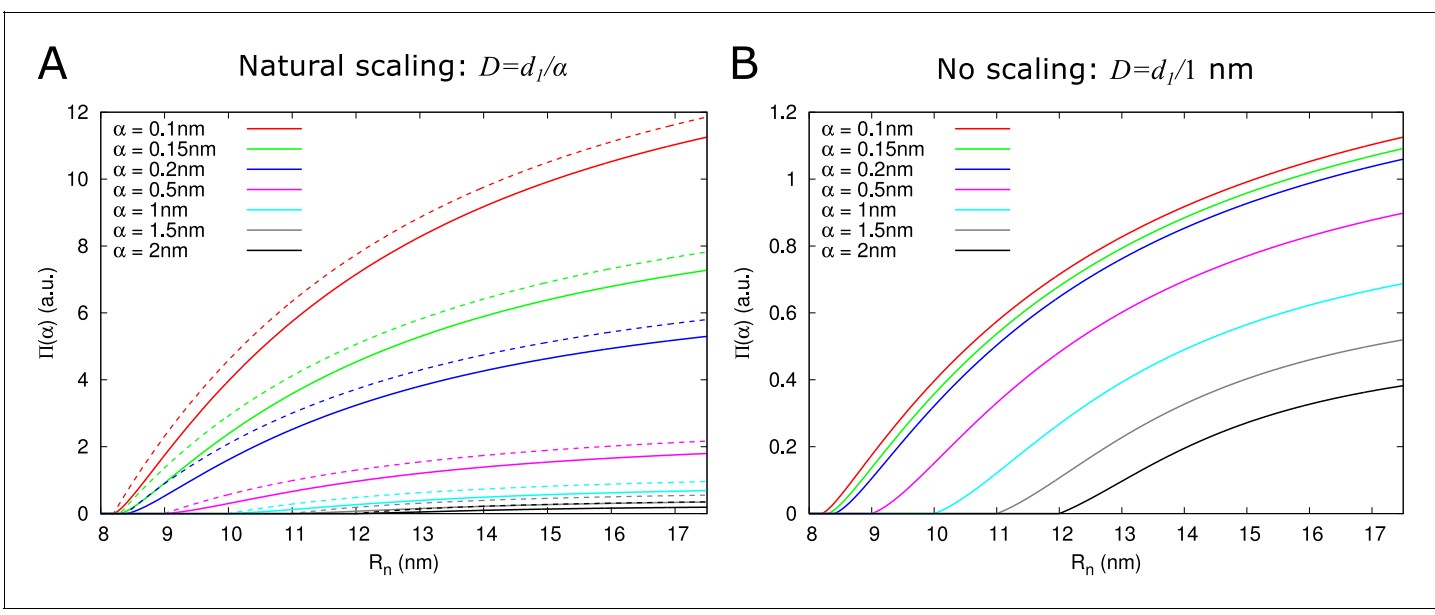

**Figure 3.** Impact of particle size (radius = $\alpha$) on single pore effective permeability $\Pi(\alpha)$. (A) Dependence of $\Pi(\alpha)$ on neck radius ($R_n$) and $\alpha$ (different line colours, see legend). The diffusion constant $D$ is inversely proportional to particle size ($D = d_1/\alpha$). Dashed lines show $\Pi(\alpha)$ considering only steric hindrance, solid lines include all hindrance effects. B: Using the same diffusion constant for all particle sizes instead shows that, once particles can pass easily, the particle size dependence of $\Pi(\alpha)$ is largely due to the relation between particle size and diffusion constant. Parameters for calculations: $l = 200$ nm, $l_n = 25$nm, $R_{dt} = 8$ nm, $R_c = 17.5$ nm. For simplicity we use $d_1 = 1$ nm$^3$/s in this figure. Therefore, only the relative values of the unit permeabilities have meaning (consequently expressed in arbitrary units [a.u.]).

The online version of this article includes the following figure supplement(s) for figure 3:

**Figure supplement 1.** Impact of hindrance effects on $\Pi(\alpha)$.

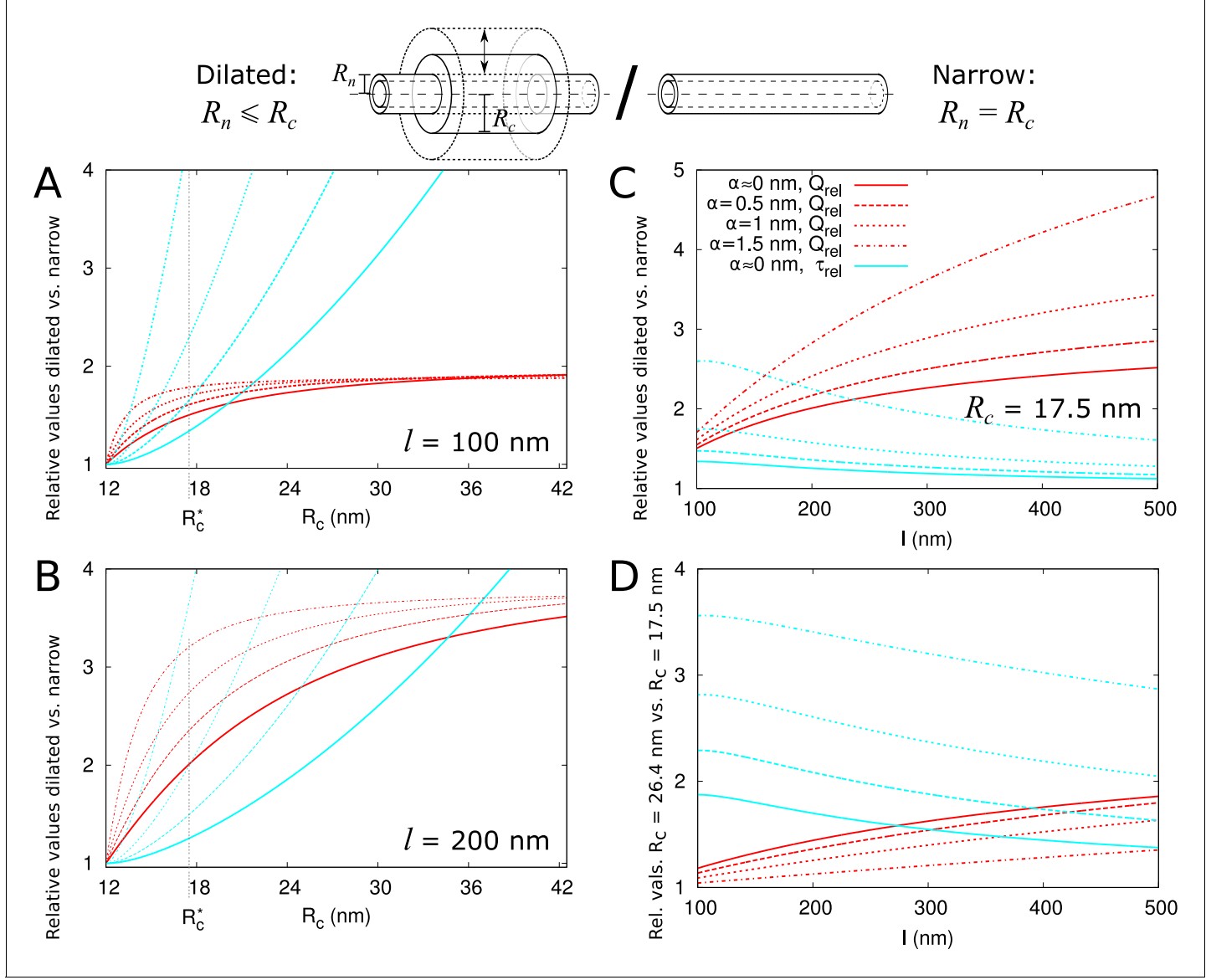

**Figure 4.** Impact of central region dilation on molar flow rate ($Q$) and mean residence time ($\tau$). The same legend shown in C applies to all panels. Narrow channels have $R_n = R_c = 12$ nm, whereas for necked/dilated channels, $R_n = 12$ nm but $R_c$ varies. (A-C) Red curves show the relation between molar flow rate in dilated PD vs narrow PD $Q_{rel} = Q_{dilated}(R_n, R_c)/Q_{narrow}(R_n)$ whereas cyan curves show the relation between mean residence time in dilated PD vs narrow PD: $\tau_{rel} = \tau_{dilated}(R_n, R_c)/\tau_{narrow}(R_n)$. Both quantities are computed for different particle sizes (solid: $\alpha \approx 0$, dashed: $\alpha = 0.5$ nm, sparse dashed: $\alpha = 1$ nm, dash-dotted: $\alpha = 1.5$ nm). (A, B) $Q_{rel}$ and $\tau_{rel}$ are shown as a function of the radius in the central region $R_c$ for different PD lengths (cell wall thickness) (A) $l = 100$ nm, (B) $l = 200$ nm. (C) Values calculated for $R_c = 17.5$ nm ($R_c^*$ in A,B) as a function of PD length. (D) Ratios of curves calculated for $R_c = 17.5$ nm (C) and $R_c = 26.4$ nm (*Figure 4—figure supplement 1B*) represented for varying PD lengths. Other parameters used for modelling are: $l_n = 25$ nm, $R_n = 12$ nm, $R_{dt} = 8$ nm.

The online version of this article includes the following figure supplement(s) for figure 4:

**Figure supplement 1.** Additional panels: $l = 500$ nm (similar to **A, B**), $R_c = 26.4$ nm (similar to **C**).

**Figure supplement 2.** Impact of neck length $l_n$ on $\Pi(\alpha)$, $Q_{rel}$ and $\tau_{rel}$.

molar flow rate increases with the central radius but quickly saturates, whereas mean resident time increases without upper bound. Moreover, both quantities increase faster for larger particle sizes ($\alpha$, dashed lines). In fact, from studying the limiting behaviour of the underlying formulas, we found that $Q_{rel}$ is always less than its theoretical maximum $\frac{l}{2l_n}$, whereas $\tau_{rel}$ ultimately scales quadratically with $R_c$, and, equivalently, linearly with the surface ratio $\tilde{A}_c/\tilde{A}_n$ (see Appendix 3 and *Appendix 3—figure 1*).

In simpler terms: the benefits of increased transport volume with increasing $R_c$ saturate, and instead the costs in transport time increases ever faster with further dilation of the central region. This defines a trade-off between transport volume and transport time with increasing $R_c$ when we analyze a single PD with a given entrance area.

Our computations also show that with increasing PD length $l$, the balance between both factors shift, because a much larger increase of $Q_{rel}$ is possible (**Figure 4A–C**). Similarly, for any given combination of $R_n$ and $R_c$, $Q_{rel}$ decreases with increasing $l_n$ and decreases faster for shorter $l$, whereas $\tau_{rel}$ has its maximum at $\tilde{l}_n = l/2$ (**Figure 4—figure supplement 2**). Together, these computations suggests that dilation of the central region is more favourable in thicker cell walls. Interestingly, this theoretical observation correlates well with a recent EM study in *Arabidopsis* root tips (**Nicolas et al., 2017b**). The authors observed that PDs with a distinct dilated central region and neck region occurred mostly in thicker cell walls (average 200 nm), whereas in thin cell walls (average 100 nm), they found mostly straight PDs.

Additionally, (**Nicolas et al., 2017b**) observed a smaller and less variable radius in channels where the central region was occupied by spokes compared to channels without them ($R_c$ = 17.6 nm vs. 26.4 nm on average). To analyze the effects of these changes on molar flow rate and MRT, we redrew the curves to compute relative values for $R_c$ = 26.4 nm and $R_c$ = 17.5 nm as a function of PD length (cell wall thickness) and for various particle sizes. As an example, panel C shows the variations observed when considering $R_c$ = 17.5 nm ($R_c^*$ in A,B). We found that the molar flow rate $Q_{rel}$ increases less than the MRT $\tau_{rel}$ when increasing $R_c$ from 17.5 nm to 26.4 nm, except for the smallest particle sizes in combination with large $l$ (**Figure 4D**). These data suggest that in cell walls of moderate thickness, restricting the radius of the central region (which can be achieved by adding spokes) improves overall performance.

In summary, transport time and transport volume scale differently with the radius of the central region thus producing PDs with a dilated central region becomes more favourable when cell wall thickness increases. However, if the radius of the central region becomes too wide (as exemplified here for $R_c$ = 26.4 nm) the increase in transport volume does not compensate for the delay in transport time. Interpretation of this result might explain why mostly straight PDs are found in recently divided cells (with thin cell walls) and why spokes (potentially limiting the radius of the central region) are often observed in mature PDs.

## For the same given maximum particle size a PD with desmotubule can transport more than a PD without

A conserved feature of PDs –at least in embryophytes– is the presence of the DT, so we asked how this structure affects the transport capacity for particles of various sizes. In our model, the DT and the neck radius jointly define the maximum particle radius $\bar{\alpha}$. Assuming that control over maximum particle size $\bar{\alpha}$ is important and a high net flux often is desirable, we estimated the number of cylindrical channels required to match a single PD with DT. Using that $P(\alpha)$ is proportional to orifice area ($\approx A_n$), we first computed $n_c(\bar{\alpha})$, the number of circular channels that would offer the same $A_n$ as a single channel with a DT of radius $R_{dt}$ = 8 nm and the same $\bar{\alpha}$:

$$n_c(\bar{\alpha}) = \frac{(R_{dt} + 2\bar{\alpha})^2 - R_{dt}^2}{\bar{\alpha}^2} = 4\frac{R_{dt} + \bar{\alpha}}{\bar{\alpha}}. \tag{7}$$

**Figure 5A** displays the $n_c(\bar{\alpha})$ as a function of the maximum particle size. As an example, when $\bar{\alpha}$ = 2 nm, 20 cylindrical channels without DT would be needed to match the orifice surface area of a single channel with DT (with $R_{dt}$ = 8 nm). This number decreases for larger $\bar{\alpha}$. We then considered that not all of this surface area is available for transport because of hindrance effects (**Figure 2B–D**). We found that even if the total surface area is the same, the channel with DT has a larger available surface area than the equivalent number of cylindrical channels. This is because in cylinders a larger fraction of the surface is close to the wall and, hence, hindrance effects are much more severe (**Figure 5B**, **Figure 5—figure supplement 1**). The difference increases with increasing relative particle size ($\alpha/\bar{\alpha}$). Steric hindrance, that is the center of a hard particle cannot come closer to the wall than its own radius, plays only a minor part in this effect (**Figure 5B**).

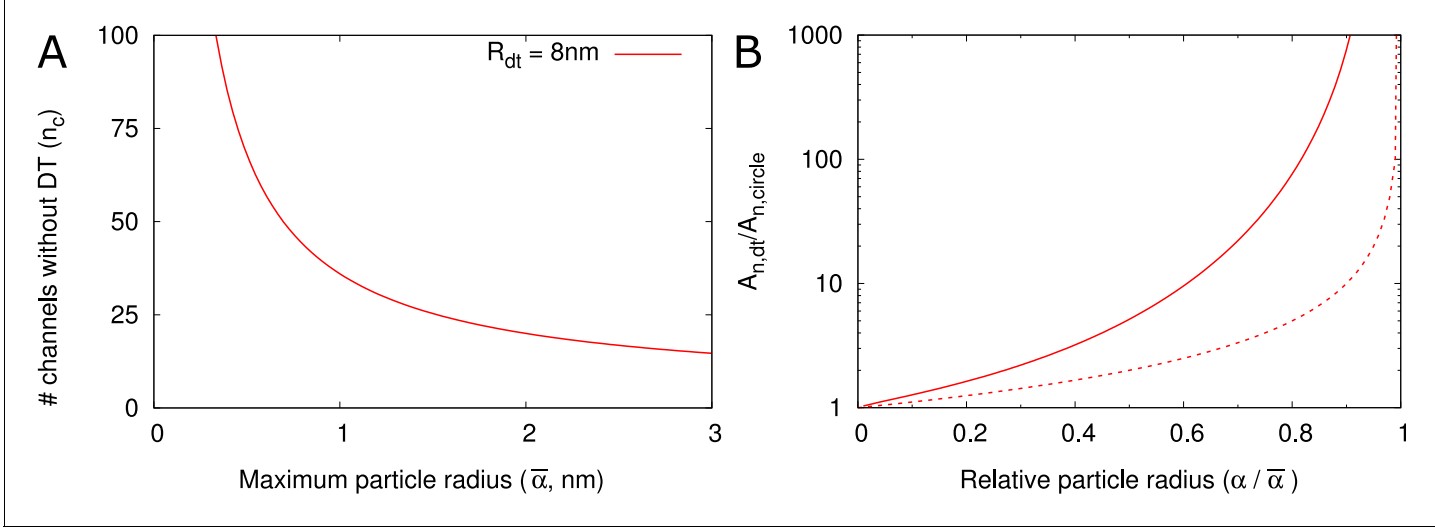

**Figure 5.** DT increases the cross section surface area available for transport per channel given a maximum particle radius $\bar{\alpha}$. (**A**) The number of cylindrical channels ($n_c$) that is required to match the total entrance surface of a single channel with $R_{dt}$ = 8 nm and the same maximum particle radius $\bar{\alpha}$. (**B**) Shows the relative area available for transport ($A_n$) in relation to relative particle size ($\alpha/\bar{\alpha}$) when comparing channels with DT and the equivalent number of cylindrical channels. Total surface area is the same. Solid lines include all hindrance effects ($\tilde{\tilde{A}}_{n,dt}/\tilde{\tilde{A}}_{n,circle}$; cf. **Figure 2D**). Dashed lines includes steric effects only ($\tilde{A}_{n,dt}/\tilde{A}_{n,circle}$; cf. **Figure 2C**).

The online version of this article includes the following figure supplement(s) for figure 5:

**Figure supplement 1.** Hindrance factors with and without DT.

## Clustering of PDs in pit fields reduces effective symplasmic permeability

The cell wall is only permeable for symplasmic transport where the PDs are. In this scenario, particles have to diffuse longer distances (on average) to reach a spot to cross the wall compared to a wall that is permeable everywhere. To account for this, we have introduced a correction factor, or 'inhomogeneity factor', $f_{ih}$ in **Equation 6** for the effective symplasmic permeability. Here, we explore how $f_{ih}$ depends on all model parameters. To calculate $f_{ih}$, we treated the cytoplasm as a homogeneous medium. This simplifying assumption is necessitated by the lack of detailed information on the cytoplasm structure and how it differs among cells. Effectively, we assumed that the obstructing effects of ER, vacuoles, etc. are similar throughout the whole cell volume and thus can be captured in a single reduced cytoplasmic diffusion constant.

First, we calculated $f_{ih}$ for isolated PDs positioned on a triangular grid in the cell wall (**Figure 6A**), as described in the Materials and methods. In **Figure 6** we presented $f_{ih}$ as a function of $R_n$ and explored its dependence on particle size $\alpha$ (**Figure 6—figure supplement 1A**), presence/absence of DT (**Figure 6—figure supplement 1A**), cell length $L$ (**Figure 6—figure supplement 1B**), density of PD $\rho$ (**B**), wall thickness $l$ (**C**) and PD distribution in the wall (**D**). We found that, provided that $R_n$ is large enough for particles to enter (as indicated by vertical cyan lines in **Figure 6—figure supplement 1A**), $f_{ih}$ is independent of cell length $L$ and particle size $\alpha$ (**Figure 6—figure supplement 1A,B**) and is not affected by the DT. We also adjusted the computation for different regular trap distributions (**Berezhkovskii et al., 2006**) to find that $f_{ih}$ also hardly depends on the precise layout of PDs (**Figure 6D**). Although variations in $f_{ih}$ appear larger at low PD densities, for typical $R_n$ values (for example, 12 nm as in **Figure 4**) density only has a minor impact (**Figure 6B**). Finally, we found an increase of $f_{ih}$ with increasing PD length $l$, saturating to its theoretical maximum of $f_{ih} = 1$ in thick cell walls ($l > 500$ nm) (**Figure 6C**). This result reflects the increasing time required for passing the PD itself with increasing PD length and, hence, a decreasing relative importance of the cytoplasmic diffusion.

Second, we investigated the effect of PDs grouped in small clusters resembling pit fields (see Materials and methods). The average centre-to-centre distance between PDs in pit fields considerably varies across species, with reported/calculated distances between 60 and 180 nm

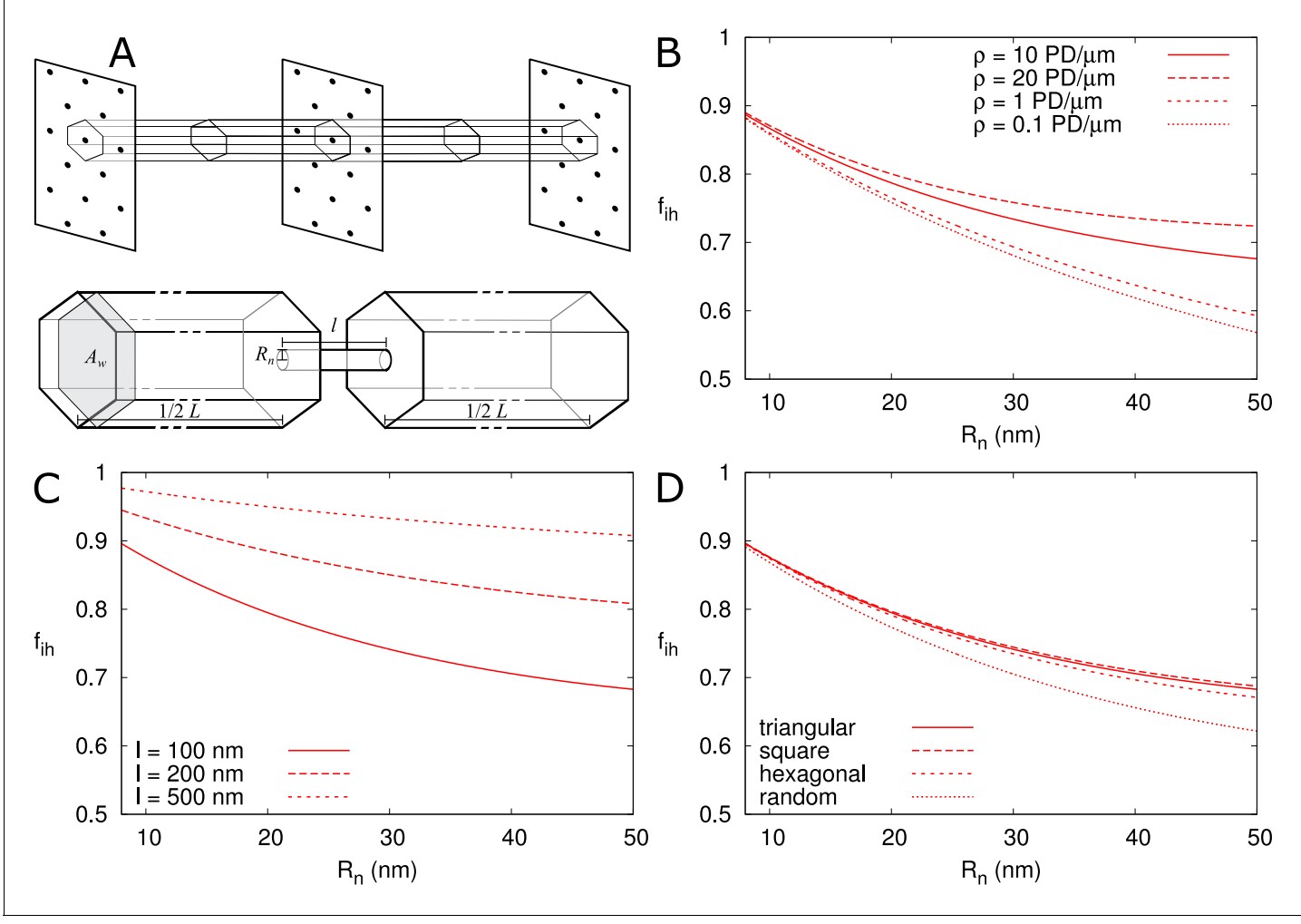

**Figure 6.** Correction factor $f_{ih}$ for inhomogeneous wall permeability depends on PD distribution, cell wall thickness and neck radius. (**A**) The cartoon shows the geometrical considerations and parameters used to model the diffusion towards PDs. Cell wall inhomogeneity is incorporated as a correction factor $f_{ih}$, $0 < f_{ih} \leq 1$, which measures the relative impact of cytoplasmic diffusion towards the locations of the PDs in the cell wall compared to reaching a wall that is weakly but homogeneously permeable (i.e., with $f_{ih} = 1$). The cytoplasm is considered homogeneous. Each bit of cytoplasm can be assigned to the PD closest to it. With PDs on a regular triangular grid, the cytoplasm belonging to a single PD, with an outer (neck) radius $R_n$, is a hexagonal column with cross section area $A_w$ and 1/2 of the cell length $L$ on either side of the wall. (**B-D**) $f_{ih}$ is represented as a function of $R_n$. The presence/absence of DT does not affect the values of $f_{ih}$ (*Figure 6—figure supplement 1A*). In all cases, solid lines correspond to: $l$ = 100 nm, $L$ = 10 µm, $\alpha$ = 0.5 nm, a PD density of $\rho$= 10 PD/µm$^2$, and PDs distributed on a triangular grid. Broken lines show the effects of changes in PD density $\rho$ (**B**), PD length $l$ (**C**) and PD distribution (**D**).

The online version of this article includes the following figure supplement(s) for figure 6:

**Figure supplement 1.** $f_{ih}$ is not affected by particle size $\alpha$, presence of DT, or cell length $L$.

(*Terauchi et al., 2015*; *Schmitz and Kühn, 1982*; *Danila et al., 2016*; *Faulkner et al., 2008*). The lowest values, however, are from brown algae, which have a different PD structure from higher plants (*Terauchi et al., 2012*). As a default, we used $d$ = 120 nm, which also coincides with measurements on electron micrographs of tobacco trichomes presented in *Faulkner et al. (2008)*. In *Figure 7A* we calculated $f_{ih}$ as a function of total PDs ('entrances') per area of cell wall for different numbers of PDs $p$ clustered in a single pit field. We found that $f_{ih}$ decreases with increasing number of PDs in a pit (and constant total PD density $\rho$). Different from isolated PDs, *Figure 7A* also reveals that, when grouped in pit fields, there is a strong dependence of $f_{ih}$ on total PD density (number of PD entrances per area of cell wall). This could be predicted from extrapolating *Figure 6B* for isolated PDs, where density dependence also increases with increasing PD radius, because cluster radii $R_{pit}$ are much larger than the largest $R_n$ used in *Figure 6B*. *Figure 7B* shows that clustering (in this

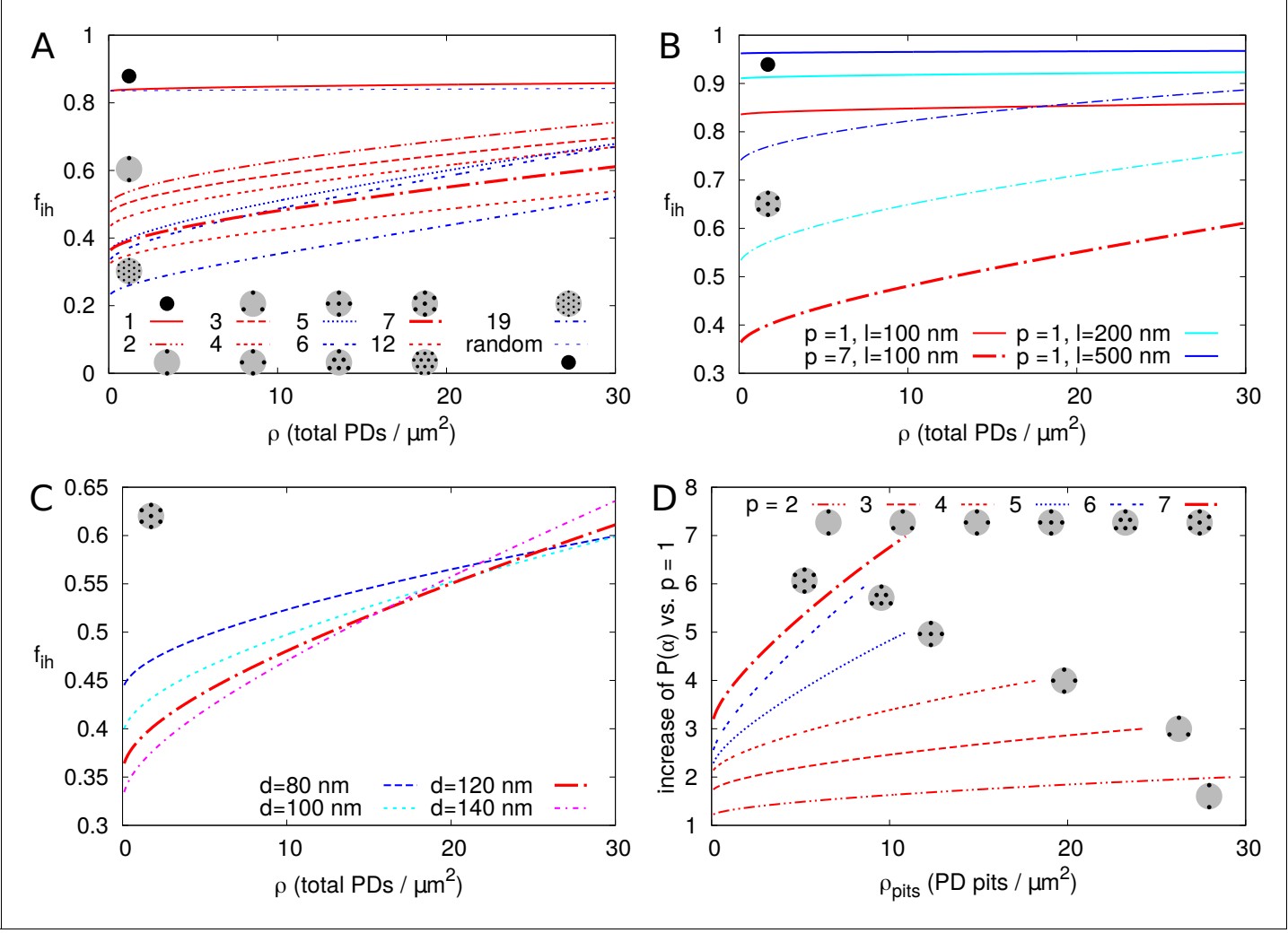

**Figure 7.** Impact of PD clustering into pit fields. PD organization within pits is indicated with small cartoons in each graph. Pits themselves are distributed on a regular triangular grid. Within pit fields, the nearest neighbour distance between PDs $d$ (120 nm by default) is independent of the number of PDs per pit field. (A-C) $f_{ih}$ is represented as a function of total PD density $\rho$ (the total number of PD entrances per unit of cell wall area) for: a varying number of PDs per cluster $p$ (as indicated by line type, (A), for different PD length $l$ (B, solid lines: isolated PDs, dash-dotted lines: 7 PDs per cluster, red colour indicates $l$: 100 nm, cyan for 200 nm, blue for 500 nm) and for different PD spacing within clusters (C, shown for clusters of 7 PDs with centre-to-centre distance $d$ as indicated by line type and colour). Cluster sizes 5, 6, and 19 are indicated with blue lines for readability (A,D). For comparison, $f_{ih}$ for non-clustered but randomly distributed PDs is also indicated. (D) The impact of increasing the number of PDs per cluster $p$ on $P(\alpha)$ as a function of cluster density $\rho_{pits}$ (the number of pit fields per unit of cell wall area). Lines show the fold increase of $P(\alpha)$ when increasing the number of PDs per cluster from one to the number indicated by the line type (same as in A). Lines are terminated where $f_{ih}$ of clusters meets $f_{ih}$ of isolated PDs at the same total PD density. Beyond that, calculation results are no longer reliable because clusters get too close and the impact of clustering on $f_{ih}$ could be considered negligible. (A-D) Default parameters: $l$ = 100 nm, $d$ = 120 nm, $R_n$ = 12 nm.

case 7 PDs) increases the dependence of $f_{ih}$ on PD length (compare solid and dashed lines of the same colour). Increasing the distance between PDs within the cluster (*Figure 7C*), also increases the dependence of $f_{ih}$ on PD density. Also the arrangement of PDs in small model clusters affects the degree of dependence $f_{ih}$ on $\rho$. In both cases, we observe the steepest dependency of $f_{ih}$ on $\rho$ for the clusters with the lowest *within cluster* PD density (pit fields with $p$ = 5, 6 and 19: indicated with blue lines in *Figure 7A*; see also *Table 1*).

It is hypothesized that PD clustering arises or increases in the process of increasing PD number post cytokinesis, possibly through (repeated) 'twinning' of existing PDs (*Faulkner et al., 2008*). We, therefore, also investigated the effect of increasing the number of PDs per cluster ($p$), starting from

**Table 1.** Pit radius ($R_{pit}$) as a function of number of PDs per pit.
The third and fourth column show numerical values for $d$ = 120 nm and $R_n$ = 12.

| PDs/pit | $R_{pit}$ | $A_{PD}/A_{pit}$ | $R_{pit}$ |
|---|---|---|---|
| 1 | $R_n$ | 1 | 12 |
| 2 | $R_n + \frac{1}{2}d$ | 0.056 | 72 |
| 3 | $R_n + \frac{1}{3}\sqrt{3}d$ | 0.065 | 81.3 |
| 4* | $R_n + \frac{1}{2}\sqrt{2}d$ | 0.061 | 96.9 |
| 5* | $R_n + d$ | 0.041 | 132 |
| 6 | $R_n + \frac{2}{3}\sqrt{3}d$ | 0.038 | 150.6 |
| 7 | $R_n + d$ | 0.058 | 132 |
| 12 | $R_n + \frac{1}{3}\sqrt{13}d$ | 0.071 | 156.2 |
| 19 | $R_n + 2d$ | 0.043 | 252 |

*: All entries are based on PDs on a triangular grid within each pit, except for 4 and 5, where the PDs inside a pit are arranged on a square grid. Clusters (pitfields) are always arranged on a triangular grid.

1 PD per cluster (*Figure 7D*). As expected, $P(\alpha)$ always increased with the increase in cluster size/PD number (*Figure 7D*), despite the decrease in $f_{ih}$ compared to homogeneously distributed PDs. This increase was larger for larger pit densities (number of pit fields per cell wall area).

In summary, for isolated and roughly evenly distributed PDs, the correction factor $f_{ih}$ for inhomogeneous wall permeability has only a minor role on $P(\alpha)$. For realistic PD dimensions ($R_n$ < 20–25 nm), the additional effect of $f_{ih}$ with parameter changes would be too small to be observed experimentally, with the possible exception of PD length $l$. However, when considering clusters of PDs, as is common in pit fields, $f_{ih}$ is markedly reduced, and PD length and density have a much larger impact on $f_{ih}$. We observed the biggest difference between isolated PDs and pairs, that is when going from single to twinned PDs (*Figure 7A*).

## Application of the model to compute effective permeability for fluorescein derivatives

In a system where non-targeted symplasmic transport is fully driven by diffusion (so no (significant) active transport or hydrodynamic flow), our calculations using reasonable PD dimensions and densities should yield values close to the ones measured experimentally. As a resource to test this hypothesis, we have build a Python program, PDinsight, that computes effective permeabilities from parameter measurements extracted from EM. As some of these parameters might be more reliable than others, we also created a mode in the program to predict what are the minimum requirements in terms of parameter (combination of parameters) values to obtain experimentally measured symplasmic permeability. Exploring these requirements is equivalent to testing hypotheses like: 'What if PD aperture is larger than observed with EM? or if the molecular radius is smaller than predicted?". Predictions made with the program can be used to explain experimental results, highlight areas/parameters that need more investigation and can help with the design of new strategies to change effective symplasmic permeability in vivo. For a full description of the program and its possibilities, see Appendix 6.

As a test case, we used the model to explain the permeability measurements in *Arabidopsis thaliana* roots reported for carboxyfluorescein (CF) diacetate: a membrane permeable non-fluorescent dye that once converted inside cells into a fluorescent version of fluorescein can only move from cell to cell via the PDs (*Rutschow et al., 2011*). Using a technique named fluorescence recovery after photobleaching (FRAP), CF effective permeability was estimated for transverse walls in the root meristem zone (measured ≈ 200 μm from the quiescent centre). The authors present two experimental setups: a 'tissue level' experiment in which a whole ≈ 50 μm longitudinal section of the root was bleached (estimated effective permeability 6–8.5 μm/s) and a single cell experiment in which a single epidermal cell was bleached (estimated effective permeability 3.3 ± 0.8 μm/s).

PD densities in transverse walls of *Arabidopsis thaliana* roots were reported by *Zhu et al. (1998)*: vascular: 9.92 ± 0.58, inner cortex: 12.28 ± 0.67, outer cortex: 9.08 ± 0.50 and epidermis 5.42 ± 0.42

PDs/$\mu m^2$. Based on these numbers we assume a PD density of 10–13 PDs/$\mu m^2$ for the tissue level experiment and 5 PDs/$\mu m^2$ for the single cell experiment. Fluorescein has a Stokes radius of approximately 0.5 nm (*Champion et al., 1995*; *Corti et al., 2008*) and a cytoplasmic diffusion constant of $D = 162 \mu m^2/s$ (one third of its water value) (*Rutschow et al., 2011*). Feeding these numbers to the model, and considering that PDs appear as straight channels in these walls (*Nicolas et al., 2017b*), we are able to reproduce the measured permeability values for observed PD densities (*Zhu et al., 1998*) only if we assume a relatively wide open neck ($R_n > 15$ nm) (*Figure 8A,B*, *Table 2*). Notably, the required neck radius for the single cell experiment fits within the range of the tissue level experiment when considering the respectively measured densities. This prediction is plausible if we consider that, in the same tissues, GFP (a protein with a reported hydrodynamic radius of 2.45 nm [*Calvert et al., 2007*] to 2.82 nm [*Terry et al., 1995*]) moves intercellularly (*Stadler et al., 2005b*). Using our default $R_{dt}$, $R_n$ should be distinctly wider than 13–14 nm for GFP to move. We also explored the possibility that PD densities are higher than determined by *Zhu et al. (1998)*. We found that to obtain the required effective permeabilities for CF with our default $R_n = 12$ nm, we would need PD densities of 33–47 PDs $\mu m^{-2}$ for the tissue level experiment and 19 (14 - 23) PDs

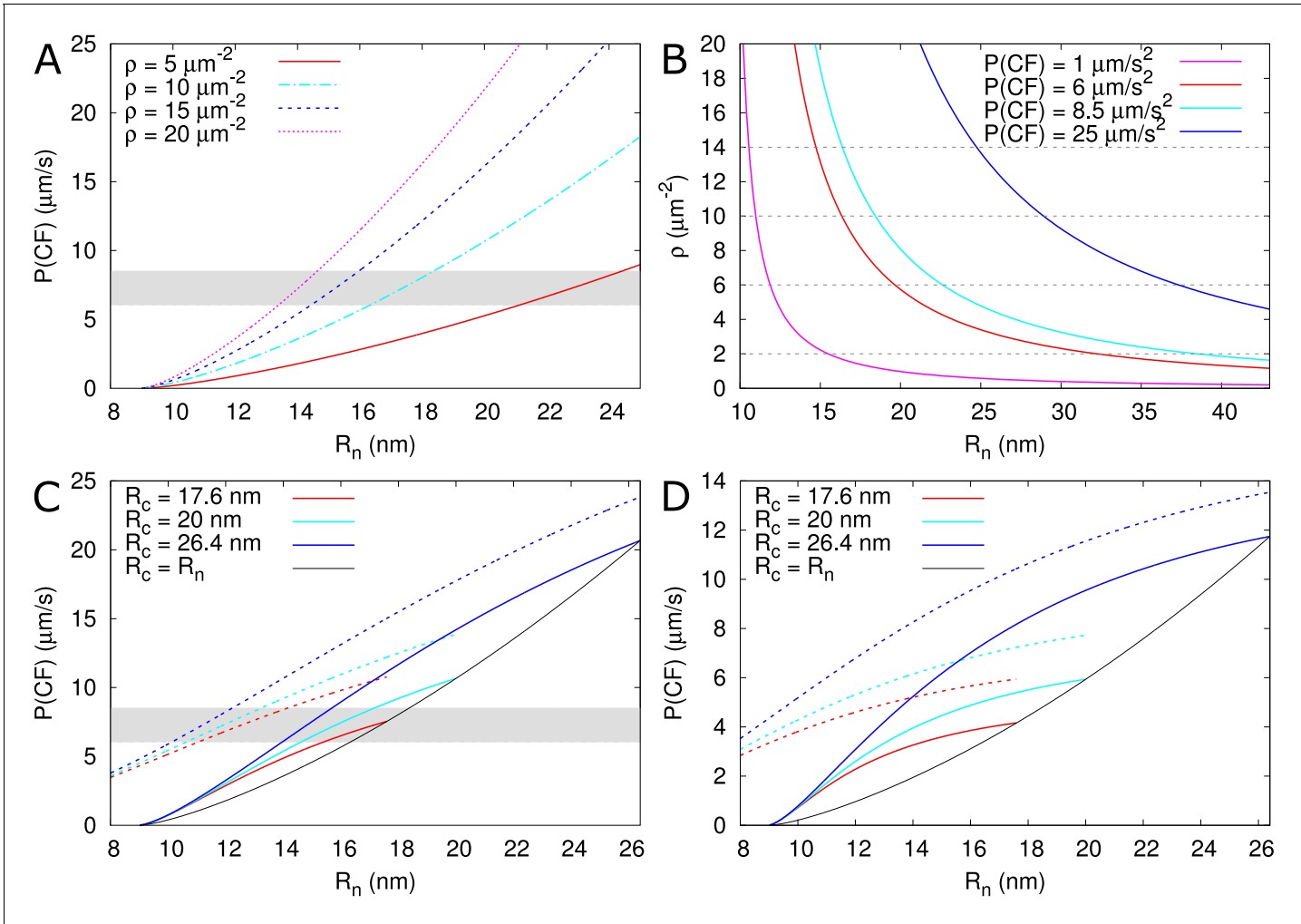

**Figure 8.** Calculated effective permeabilities for carboxyfluorescein (CF) as a function of PD aperture at the neck $R_n$. (A, B) Shows the graphs for straight channels. (A) Effective permeabilities are calculated for different PD densities (different colour curves). The horizontal gray band in A and C indicates the cortical values observed by *Rutschow et al. (2011)*. (B) Shows the PD density required to obtain measured values of $P(CF)$ (different colour curves) as a function of $R_n$. Horizontal broken lines are introduced to aid readability. (C, D) Shows that effective permeability increases with dilation of the central region ($R_c > R_n$). As a reference, values for straight channels are indicated in black. Dashed curves show values calculated for channels without DT. (D) Shows the same calculations as C but for longer PDs $l = 200$ nm. Default parameters: $\alpha = 0.5$ nm, $D = 162 \mu m^2/s$, $l_n = 25$ nm, $l = 100$ nm, $R_{dt} = 8$ nm, $\rho = 10$ PD/$\mu m^2$, PDs are spaced on a triangular grid, without clustering.

**Table 2.** Parameter requirements for reproducing measured $P(CF)$ values (***Rutschow et al., 2011***) with the default model.

This table was generated using PDinsight. A: Required density ($\rho$) for a given $\bar{\alpha}$ and neck radius ($R_n$). B: Required $\bar{\alpha}$ and corresponding $R_n$ for a given $\rho$. C: values required to reproduce $P(CF) = 25$ µm/s. Values computed for a 2x, 3x and 4x increase of $\rho$ are also shown. This is done both for a uniform increase of the density ($p = 1$) and for (repeated) twinning ($p>1$) from a uniform starting density (indicated in bold). $p$ is the number of PDs per pit.

| A $P(CF)$(µm/s) | $\bar{\alpha}$ (nm) | $R_n$ (nm) | $\rho$ (PD/µm$^2$) |
|---|---|---|---|
| 3.3* | 2.0 | 12 | 18.6 |
| | 2.5 | 13 | 12.6 |
| | 3.0 | 14 | 9.3 |
| | 3.4 | 14.8 | 7.6 |
| | 3.5 | 15 | 7.3 |
| | 4.0 | 16 | 5.9 |
| 6 | 2.0 | 12 | 33.5 |
| | 2.5 | 13 | 22.7 |
| | 3.0 | 14 | 16.8 |
| | 3.4 | 14.8 | 13.8 |
| | 3.5 | 15 | 13.2 |
| | 4.0 | 16 | 10.7 |
| 8.5 | 2.0 | 12 | 47.2 |
| | 2.5 | 13 | 32.0 |
| | 3.0 | 14 | 23.7 |
| | 3.4 | 14.8 | 19.4 |
| | 3.5 | 15 | 18.5 |
| | 4.0 | 16 | 15.0 |

| B $P(CF)$(µm/s) | $\rho$ | $\bar{\alpha}$ | $R_n$ (nm) |
|---|---|---|---|
| 3.3* | 5 | 4.5 | 16.9 |
| | | 4.2 | 16.5 |
| 6 | 10 | 4.2 | 16.3 |
| | 13 | 3.5 | 15.1 |
| 8.5 | 10 | 5.2 | 18.4 |
| | 13 | 4.4 | 16.8 |
| 1 | 10 | 1.5 | 11.0 |
| | 13 | 1.3 | 10.6 |

| C $P(CF)$(µm/s) | $\rho$ | $p$ | $\bar{\alpha}$ | $R_n$ (nm) |
|---|---|---|---|---|
| 25 | 10 | 1 | 10.5 | 28.9 |
| | 20 | 1 | 6.6 | 21.2 |
| | | 2 | 7.2 | 22.5 |
| | 30 | 1 | 5.1 | 18.1 |
| | | 3 | 5.6 | 19.2 |
| | 40 | 1 | 4.2 | 16.4 |
| | | 4 | 4.6 | 17.2 |
| | 13 | 1 | 8.8 | 25.6 |
| | 26 | 1 | 5.6 | 19.1 |
| | | 2 | 6.0 | 20.0 |
| | 39 | 1 | 4.3 | 16.6 |

*Table 2 continued on next page*

|  |  |  |  |
|---|---|---|---|
|  |  | 3 | 4.6 | 17.2 |
| 52 | 1 | 3.6 | 15.1 |
|  |  | 4 | 3.8 | 15.5 |

*: Single cell experiment. All other data relates to tissue level experiments.

$\mu m^{-2}$ for the single cell experiment (**Table 2**). The ratio of these required densities is in line with the observed ratio of relevant densities (**Zhu et al., 1998**).

Using the model, we also explored the effect of 'necked'/'dilated' PDs by adding a wider central region to PDs. For a central radius $R_c$ = 17.6 nm, the required $R_n$ to reproduce the tissue level CF permeability values would decrease by perhaps 1 nm or at most 3 nm (for $R_c$ = 26.4 nm) considering a PD density in the order of $\rho$ = 10 $\mu m^{-2}$ (**Figure 8C**, $R_c$ values from **Nicolas et al., 2017b**). In thicker cell walls (**Figure 8D**), the calculated effective permeabilities increased relatively more, but remained too low, suggesting that increasing cavity radius is never sufficient for reproducing the **Rutschow et al. (2011)** values (see also **Figure 4**).

Using the tissue level setup, Rutschow et al. also reported drastic changes in effective permeability after $H_2O_2$ treatment. They found a strong decrease in symplasmic permeability to $\approx$ 1$\mu m$/s after treatment with a 'high' $H_2O_2$ concentration, which was explained by rapid PD closure through callose deposition. Using our program we found that, for this reduction of $P(CF)$, callose must reduce $R_n$ to 11 nm ($\rho$ = 10 $\mu m^{-2}$) or 10.6 nm ($\rho$ = 13 $\mu m^{-2}$), resulting in $\bar{\alpha}$ = 1.5 nm or 1.3 nm, respectively. The authors also found a strong increase in permeability to $\approx$ 25 $\mu m$/s after treatment with a 'low' $H_2O_2$ concentration. Reproducing this increase requires a large change at the PD level. At the extremes, an increase of $R_n$ to approximately 29 nm for $\rho$ = 10 $\mu m^{-2}$ (**Figure 8A,B**, **Table 2B**), or a slightly more than four fold increase in PD density would be required to reproduce this high effective permeability (**Table 2C**). Alternatively, both $R_n$ and $\rho$ would have to increase substantially (**Figure 8B**). As an extreme hypothesis, we also calculated the effects of complete DT removal. The increases in $P(\alpha)$ that could be obtained this way were by far insufficient to explain the reported effect of mild $H_2O_2$ treatment (**Figure 8C,D**), making DT modification or removal a highly unlikely explanation for this change.

Taken together, these calculations indicate that our model for diffusive symplasmic transport can indeed explain experimentally observed measurements of effective symplasmic permeability, but only with somewhat wider PDs/neck regions than expected yet in line with the observed permeability for GFP and within the range of PD diameters measured in thick cell walls. Alternatively, similar changes in symplasmic permeability can be achieved with several fold higher densities than typically measured. These predictions provide a framework for experimental validation. We also compared the results obtained with our unobstructed sleeve model and the sub-nano channel architecture. Using the sub-nanochannel architecture, much larger PD densities would be required to achieve the same $P(CF)$: roughly twice as large for $\bar{\alpha}$ = 3.5–4 nm and even larger for smaller $\bar{\alpha}$ (see Appendix 5 and **Appendix 5—table 1**). These results favour unobstructed sleeve models for offering more plausible hypotheses to explain the experimental results for CF and the impact of $H_2O_2$ treatment on effective permeability.

## Discussion

In this manuscript, we presented a method for calculating effective wall permeabilities for non-targeted, diffusive symplasmic transport based on the dimensions and distribution of PDs and on the size of the mobile particles. For individual PDs, we used a minimal geometrical description that allowed us to extensively investigate the effects of dilation of the central PD region and the implications of a DT at the PD axis on transport properties. Because PDs are narrow, our calculated effective symplasmic permeabilities were heavily affected by molecular hindrance effects. For the effects of PD distribution, we introduced an 'inhomogeneity factor' $f_{ih}$ between 0 and 1, which accounts for the reduction in overall permeability due to spatial arrangement of PDs. We found that the degree

of PD clustering had a strong impact on this factor, whereas the exact spatial distribution of either isolated PDs or clusters had little impact.

Our model uses an unobstructed cytosolic sleeve for symplasmic transport. In such models, the DT gives the PD an annular cross section, which strongly increases transport capacity compared to cylindrical channels with the same $\bar{\alpha}$ and total cross section area at the entrance, particularly for relatively large molecules. Having a DT offers an additional flexibility in regulating size selectivity through the possibility of a dilated state of the PD by displacement or temporal removal of the DT (*Zambryski and Crawford, 2000*; *Crawford and Zambryski, 2000*). This feature, however, can be exploited for the spreading of viruses (*Benitez-Alfonso et al., 2010*) and other intracellular parasites such as the fungus *Magnaporthe oryzae* (*Kankanala et al., 2007*). Functional PDs without DT (and inner diameter of 10–20 nm) have been reported for the brown algae species *Dictyota dichotoma* (*Terauchi et al., 2012*). Due to their very high membrane curvature, DT formation requires curvature-inducing proteins (such as reticulons) and a special lipid composition (*Tilsner et al., 2011*; *Grison et al., 2015*; *Knox et al., 2015*). It is likely that performance benefits of the DT offset these costs and disadvantages and it is therefore under evolutionary selection. Additionally, the connection between DT and ER could result in variable degrees of PD occlusion and hence a potential control mechanism for PD accessibility. *Park et al. (2019)* have started to explore this concept in the context of pressure regulated PD occlusion.

We have also calculated the performance costs (transport rate) and benefits (transport volume per PD) of having distinct central and neck regions. Whereas the transport time scales quadratically with the radius of the central region ($R_c$), the relative transport volume has a strong upper bound that increases with channel length. These results suggest that straight PDs perform better in thin (average 100 nm) cell walls and necked/dilated PDs in thick (average 200 nm or more) cell walls, which correlates with recent observations (*Nicolas et al., 2017b*). This is not, however, the only way to explain these observations. Necked/dilated PDs might appear because (1) size selectivity is more efficiently controlled by restricting callose deposition to a 20–30 nm long neck region, (2) the formation of 'spokes' in the central region leads to this narrow-wide-narrow structure, and/or (3) the material properties of cellulosic cell walls and PD cell membranes only allow for a distinctly wider central region if the channel is long enough.

In our model, we naturally define the SEL as $\bar{\alpha}$, the maximum particle radius that could fit through the model PD, but experimental determination of this value is difficult and often relies on the transport of detectable, typically fluorescent, molecules such as CF. The limited set of suitable molecules, particularly for non-invasive techniques, introduces a large uncertainty in SEL measurements and hence $\bar{\alpha}$. Also other biological factors could lead to an underestimation as well as an overestimation of $\bar{\alpha}$. For example, in so-called active symplasmic phloem loaders, such as the cucurbits, sucrose moves symplasmically from bundle sheet cells (BSC) to intermediary cells (IC), where it is polymerized into the larger oligomers raffinose and stachyose, that do not diffuse back in detectable amounts (*Haritatos et al., 1996*; *Liesche and Patrick, 2017*). Two explanations have been suggested: (1) a discriminating PD SEL at this interface, which prevents the back transport of raffinose and stachyose (*Liesche and Schulz, 2013*), or (2) open PDs combined with a directional flow which could be sustained by the xylem flow (*Comtet et al., 2017*). Only the latter could explain the observed amount of sucrose transport (*Liesche and Schulz, 2013*; *Comtet et al., 2017*). This example illustrates that the consideration of a symplasmic flow could largely affect calculated permeabilities and fluxes.

An overestimation of $\bar{\alpha}$ could occur for non-spherical molecules or temporal variations in PD properties. Although a molecule's hydrodynamic radius is a better predictor of its symplasmic transport efficiency than its molecular weight (*Terry and Robards, 1987*; *Dashevskaya et al., 2008*), it conceptually assumes a static replacement sphere. Molecules may be more flexible and/or have a shortest dimension than what is captured by its diffusive behaviour in bulk. PDs might also accommodate molecules that are larger than expected, either through interactions with specific PD proteins (*Benitez-Alfonso et al., 2010*) or because membranes and/or cell wall domains around PDs allow for reversible transient modifications in $\bar{\alpha}$ (*Abou-Saleh et al., 2018*). Additionally, molecules could pass in the wake of larger proteins/complexes/structures that modify PDs (e.g., tubule-forming viruses; *Amari et al., 2010*). Assessing the extent and time scales of temporal variations in PD boundaries and their implications remains an open topic for future investigation.

The framework we have developed for so-called 'simple' PDs also provides an intuition for the functional implications of complex geometries such as 'twinned', 'branched' or 'funnel' PDs (*Ehlers and Kollmann, 2001*; *Ehlers and van Bel, 2010*; *Faulkner et al., 2008*; *Ross-Elliott et al., 2017*). All else remaining equal, 'twinned' PDs have twice the entrance surface area, which would result in doubling the effective permeability $P(\alpha)$. This increase, however, will be reduced because of the less uniform PD spacing in a density dependent manner (*Figure 7A*). 'Branched' or 'complex' PDs contain multiple sub-channels (branches) on at least one side with typically a single shared central cavity connecting all branches (*Oparka et al., 1999*; *Roberts et al., 2001*; *Fitzgibbon et al., 2013*). In the leaf sink/source transition, massive branching is observed and, coincidentally, the number of PDs is reduced (*Roberts et al., 2001*). The formation of many channels per PD could help to maintain sufficient transport capacity for smaller molecules. If so, the increase in the number of typically narrower channels should be much larger than the decrease in total (simple or complex) PD number. Our computations of $f_{ih}$ after twinning suggest that minimizing the distance between sub-channels could be favourable at low to moderate PD densities (*Figure 7C*). 'Funnel' PDs are reported in tissues surrounding the phloem at the root unloading zone (*Ross-Elliott et al., 2017*) and show a wide opening on the PSE (protophloem sieve element) side and a narrow opening on the PPP (phloem pole pericycle) side. (*Ross-Elliott et al., 2017*) model these as a triangular funnel that reaches its narrowest diameter only at the (PPP) bottom. There appears to be, however, a longer neck-like region at the narrow end of variable length. As hindrance is by far the highest in the narrowest section, the length of this narrow part would be a vital parameter in correctly estimating the transport permeabilities of these PDs.

We have applied our model to calculate the effective permeability for fluorescein in transverse walls of Arabidopsis root tip cells (*Rutschow et al., 2011*). Assuming purely diffusive transport and parameters based on various ultrastructural measurements, we were able to reproduce the observed effective permeabilities for CF and to assess the plausibility of different hypotheses aimed at resolving the conundrum of apparently incompatible measurements at different scales. For resolving this conundrum, we assumed that not all PD dimensions are reliably measured with EM. We could reproduce the measured values with somewhat wider PDs/neck regions or several fold higher PD densities than usually measured by EM. Of these, the increased radius seems the more plausible scenario, in line with the requirements for efficient GFP transport reported to occur among root meristem cells (*Benitez-Alfonso et al., 2009*; *Benitez-Alfonso et al., 2013*; *Nicolas et al., 2017b*), and similar to $R_c$ values reported in thicker cell walls (*Zhu et al., 1998*; *Grison et al., 2015*; *Nicolas et al., 2017b*). Remarkably, our model predicts very similar PD aperture in the transverse walls of the epidermis and the more interior root layers when considering the $\approx$ 2-fold difference in PD density (*Zhu et al., 1998*). The obvious next step would be testing more data sets of different interfaces/plant species where purely diffusive symplasmic transport is expected. First of all, it would be ideal to test if a near or complete match between tissue level and ultrastructural measurements can be produced if all measurements are performed on the same system with the same growth/treatment conditions. Additionally, more testing could yield a better understanding of potential systematic side-effects of modern EM preparation techniques and/or uncertainties in the tissue level measurements, which would show as systematic vs random required adjustments of the model parameters. A very exciting outcome would be the discovery of distinct clusters in required parameter adjustments that could be related to cell wall properties, PD or interface type, etc. Additional model testing would become easier if the results of tissue level experiments are reported in the form of effective symplasmic wall permeabilities (in μm/s), or clearly provide all information required to transform into such units.

We also used our model to predict the PD changes after treatment with high and low concentrations of $H_2O_2$ in *Rutschow et al. (2011)*. The reduced permeability after high $H_2O_2$ treatment could easily be explained by a redox induced stress response and corresponding reduction of PD aperture (e.g., at a density of 10 PD/μm$^2$, a reduction from $\bar{\alpha}$ = 4.2–5.2 nm to $\bar{\alpha}$ = 1.5 nm would be required, see *Table 2B*). The strongly increased permeability after low $H_2O_2$ treatment, however, is harder to explain. With a single parameter change, the model predicts either a very wide PD aperture of $\bar{\alpha}$ = 8.8–10.5 nm, or a ±4-fold increase in PD density (possibly through 2 rounds of twinning/duplication), or less extreme changes if both parameters increase simultaneously (see *Table 2C*). The required increase in PD density should occur relatively fast, that is within the applied incubation period of 2 hr, and is so large that it should be readily detectable with EM.

The fact is that to reproduce experimentally measured CF effective permeabilities with our model, we had to deviate from ultrastructural based values for at least one parameter. Potential sources for these variations are: (1) ultrastructural studies might underestimate $R_n$ because plants could respond to pre-EM manipulation by closing PDs, like they do in response to microinjection or particle bombardment (*Haywood et al., 2002*; *Liesche and Schulz, 2012*), (2) PD integrity could be affected during processing for TEM leading to an underestimation of PD densities, (3) the mechanical properties of cell walls and membranes provide a flexibility in the channel that could to some degree accommodate molecules larger than the apparent $R_n$ (*Abou-Saleh et al., 2018*; *Yan et al., 2019*; *Amsbury and Benitez-Alfonso, 2019*). For a passive transport mechanism, the elastic energy required for these reversible deformations would have to be in the order of a few $k_B T$ or less. A model with flexible PD lining would be required to investigate the physical limits of this 'flexibility hypothesis', which is quite an increase in model complexity compared to the hard walls used in all current models, including ours. Finally, technical issues limit the accuracy of the CF effective permeability measurements themselves, for example, the speed of confocal microscopy bounds the spatial and temporal resolution at which CF concentrations can be monitored during and after bleaching/photoactivation (*Rutschow et al., 2011*; *Liesche and Schulz, 2012*).

To assess the impact on effective symplasmic permeability of various PD distributions, including clustering into pit fields, we introduced the inhomogeneity factor $f_{ih}$ that accounts for the fact that the wall is only permeable at certain spots (i.e., where the PDs are located). Clustering into pit fields had by far the largest impact on this factor, particularly for lower PD densities. This means that not only total PD density, but also the degree of clustering is important information for calculating effective wall permeability from experimental data. The above inhomogeneity factor and the possibility of a dilated central region set our model apart from other models based on the unobstructed sleeve architecture (*Bret-Harte and Silk, 1994*; *Liesche and Schulz, 2013*; *Dölger et al., 2014*; *Ross-Elliott et al., 2017*). Using typical PD dimensions and no clustering, inhomogeneity factor $f_{ih}$ would reduce the effective symplasmic permeability by about 15%, meaning that our model would require slightly wider or more PDs to explain the same tissue level experiments with straight channels compared to the above models.

A dilated central region is also considered in *Blake (1978)*, who investigates hydrodynamic flow only. There is, however, an interesting similarity between both conditions: in both cases the driving gradient is steepest in the (narrowest part of) the neck region, be it the concentration gradient (*Appendix 2—figure 2A*) or the pressure gradient (*Blake, 1978*). When it comes to describing the PD geometry, (*Blake, 1978*), makes the opposite choice compared to us. He glues together $\sin^2$ functions with a straight middle part, resulting in a mathematically nice (i.e., continuous differentiable) function, but consequently, neck shape cannot be controlled, and neck length and the length of the widening region are linked. We, on the other hand, use an instantaneous increase in PD radius, which introduces a mild systematic error in our estimates of effective symplasmic permeability $P(\alpha)$ (Appendix 2), but results in parameters that are directly measurable on EM images.

Comparing the unobstructed sleeve architecture to the sub-nano channel architecture, we found that the latter requires roughly twice as high PD densities to produce the same permeability values $P(CF)$ in the (*Rutschow et al., 2011*) experiments. This difference is due to the increased hindrance effects in cylindrical channels vs annular channels with the same cross sectional area. In the future, sleeve models could be refined with the consideration of central spokes (*Ding et al., 1992*; *Nicolas et al., 2017b*) and variability of PD dimensions within a single cell wall (*Nicolas et al., 2017b*; *Yan et al., 2019*). Simple considerations of the available volume suggest that the addition of spokes will increase hindrance effects, but most likely to a lesser extend than the sub-nano channel structure. Detailed molecular simulations could be a valuable tool to assess this effect.

Other future applications could be the coupling of our detailed PD level calculations of effective symplasmic permeability with tissue level models, which would allow for investigating the impact of microscopic changes on developmental and physiological processes (for example see *Foster and Miklavcic, 2017*; *Couvreur et al., 2018*). Depending on the context, it would then be useful or even required to also implement hydrodynamic flow through the PDs. Many ingredients are available for doing this while maintaining the distinguishing features of our mode, including hindrance factors (*Dechadilok and Deen, 2006*), but as far as we know, the theoretical and numerical results that we use for calculating $f_{ih}$ are only available for diffusion processes, and not yet for advection.

Additionally, one may need to replace the abrupt change in PD radius by a more gradual function. The importance of this final change could be estimated using numerical simulations.

Technological advances have started to be applied for more refined determinations on ultrastructural parameters. New fixation and sectioning techniques and new technologies such as electron-tomography (ET) and Correlative Fluorescence Electron Microscopy (CFEM) are now part of the systematic study of PD connections in different plant cells, tissues and organs. In parallel, new information on structural features characterizing PDs in different plant species/developmental stages as well as on the factors controlling PD structure and function (and thereby the effective permeability of specific molecules in different developmental or environmental conditions) are emerging. Combined with this significant experimental progress, our calculations provide a functional interpretation to characteristic PD morphological features and provide a framework to investigate how transport properties depend on these ultrastructural features and particle size in the context of simple and complex PD geometries. Another level of predictive power could be unlocked by integrating our framework into larger models at the tissue to whole organism level. This opens new avenues for exploring how developmental regulation of symplasmic transport interacts with various other pathways for long and short range intercellular communication.

## Materials and methods

### Diffusive flux through a single PD

Similar to *Smith (1986)*, we assumed the flux is distributed homogeneously within each cross section along the axis of the channel. This results in a simple mapping to a 1D channel, that is that the average local flux (per unit area of cross section) ~ 1/available cross section surface. This assumption does not hold close to the transition between neck and central region, that is a sharp transition between narrow and wide cylinders. Numerical simulations showed, however, that the error introduced by the assumption of homogeneous flux turned out to less than 4 percent for $l = 200$ nm, the shortest $l$ with experimentally observed neck region in *Nicolas et al. (2017b)* (*Appendix 2—figure 1*) and will be less for longer channels. This error can be considered irrelevant given the quality of available data on PD dimensions and the many molecular aspects of PD functioning that are necessarily neglected in a simple model.

### Hindrance factors

Hindrance factors $H(\lambda)$ including both steric and hydrodynamic effects are modelled using the numerical approximations in *Dechadilok and Deen (2006)*. They present functions for cylindrical and slit pores. For PDs with a desmotubule, we use the function calculated for straight slits.

$$H(\lambda) = 1 + \frac{9}{16}\lambda \ln(\lambda) - 1.19358\lambda + 0.4285\lambda^3 - 0.3192\lambda^4 + 0.08428\lambda^5. \qquad (8)$$

This choice is supported by the steric hindrance prefactor that is included in $H(\lambda)$ (*Dechadilok and Deen, 2006*). This $\Phi(\lambda) = 1 - \lambda$ is the same as the ratio of available to full surface area $\tilde{A}_x(\alpha)/A_x$. For cylindrical channels, that is reference channels in *Figure 5* and the regular PDs after DT removal, we use

$$\begin{aligned} H_c(\lambda) = \quad & 1 + \frac{9}{8}\lambda \log(\lambda) - 1.56034\lambda + 0.528155\lambda^2 + 1.91521\lambda^3 - 2.81903\lambda^4 \\ & + 0.270788\lambda^5 + 1.10115\lambda^6 - 0.435933\lambda^7 \end{aligned} \qquad (9)$$

for $\lambda < 0.95$ and the asymptotic approximation by *Mavrovouniotis and Brenner (1988)*,

$$H_c(\lambda) = (1-\lambda)^2 \cdot \left(0.984\left(\frac{1-\lambda}{\lambda}\right)^{\frac{5}{2}}\right) \qquad (10)$$

otherwise, as suggested by *Dechadilok and Deen (2006)*.

## Relative molar flow rate and MRT

For assessing the impact of the neck constriction on PD transport, we defined two relative quantities: $Q_{rel} = Q_{dilated}/Q_{narrow}$ and $\tau_{rel} = \tau_{dilated}/\tau_{narrow}$ (*Figure 4*, *Appendix 3—figure 1*). Using *Equation 2* for $Q(\alpha)$, $Q_{rel}$ is well defined:

$$Q_{rel}(\alpha, R_c) = \frac{l\tilde{\tilde{A}}_c}{2(\tilde{l}_n)\tilde{\tilde{A}}_c + (l - 2\tilde{l}_n)\tilde{\tilde{A}}_n} \tag{11}$$

$$\lim_{R_c \to \infty} Q_{rel}(\alpha, R_c) = \frac{l}{2\tilde{l}_n} \tag{12}$$

For $\tau_{rel}$ we first needed an expression for $\tau$ itself. Ideally, this would be a MFPT, which could calculated in a way similar to $\tau_\parallel$ in the calculation of $f_{ih}$, using a narrow-wide-narrow setup. These calculations, however, critically depend on trapping rates at the narrow-wide transitions. We do not have an expression for these, because the DT takes up the central space of the channel, which, contrary to the case of $f_{ih}$, substantially alters the problem and the circular trap based calculations would result in an underestimation of the MFPT. Instead, we stuck to the homogeneous flux assumption also used for $Q(\alpha)$ and defined $\tau$ as the corresponding estimate for the mean residence time (MRT) in the channel (see *Equation 5*). Elaborating *Equation 5*:

$$\tau(\alpha) = \frac{C_l + C_0}{2D\Delta C} \frac{(2\tilde{l}_n\tilde{\tilde{A}}_n + (l - 2\tilde{l}_n)\tilde{\tilde{A}}_c)(2\tilde{l}_n\tilde{\tilde{A}}_c + (l - 2\tilde{l}_n)\tilde{\tilde{A}}_n)}{\tilde{\tilde{A}}_n\tilde{\tilde{A}}_c} \tag{13}$$

$$= \frac{C_l + C_0}{2D\Delta C}\left(4\tilde{l}_n^2 + (l - 2\tilde{l}_n)^2 + 2\tilde{l}_n(l - 2\tilde{l}_n)\left(\frac{\tilde{\tilde{A}}_c}{\tilde{\tilde{A}}_n} + \frac{\tilde{\tilde{A}}_n}{\tilde{\tilde{A}}_c}\right)\right). \tag{14}$$

Unfortunately, this depends on the concentration difference over the channel. We are interested, however, in how the MRT changes with increasing $R_c$. In our definition of $\tau_{rel}$, the concentration difference cancels from the equation, solving the problem:

$$\tau_{rel}(\alpha, R_c) = \frac{1}{l^2}\left(4\tilde{l}_n^2 + (l - 2\tilde{l}_n)^2 + 2\tilde{l}_n(l - 2\tilde{l}_n)\left(\frac{\tilde{\tilde{A}}_c}{\tilde{\tilde{A}}_n} + \frac{\tilde{\tilde{A}}_n}{\tilde{\tilde{A}}_c}\right)\right). \tag{15}$$

This method of computing $\tau_{rel}$ again depends on the homogeneous flux assumption. For an estimate of the error introduced by this approach, see Appendix 2.

## Flow towards PDs: correction for inhomogeneity of the wall permeability

To compute $f_{ih}$, we consider a linear chain of cells that are symplasmically connected over their transverse walls (*Figure 1*). We first compute mean first passage time (MFPT) $\tau_\parallel$ through a simplified PD and a column of cytoplasm surrounding it. We then convert $\tau_\parallel$ to an effective wall permeability and compare the result with the uncorrected effected permeability computed using *Equation 6* for the simplified PD geometry and $f_{ih} = 1$.

As a simplified PD, we use a narrow cylindrical channel of length $l$ and radius $R_n$, that is initially without DT. We assume that PDs are regularly spaced on a triangular grid. Consequently, the domain of cytoplasm belonging to each PD is a hexagonal column of length $L$, the length of the cell (*Figure 6*). We adjust the results reported by *Makhnovskii et al. (2010)* for cylindrical tubes with alternating diameter by changing the wide cylinder of radius $R_w$ with a hexagonal column with cross section area $A_w = 1/\rho$ and considering hindrance effects. Makhnovskii et al. use a setup with an absorbing plane in the middle of a wide section and a reflecting plane, where also the initial source is located, in the middle of the next wide section. Assuming equal diffusion constants in both sections, they report the following MFPT from plane to plane:

$$\tau_\parallel = \frac{1}{2D}\left[L^2 + l^2 + 2D\left(\frac{l}{\kappa_n} + \frac{L}{\kappa_w}\right) + lL\left(\frac{\kappa_w}{\kappa_n} + \frac{\kappa_n}{\kappa_w}\right)\right], \tag{16}$$

where

$$\kappa_w = \frac{4DR_n f\left(\frac{R_n^2}{R_w^2}\right)}{\pi R_w^2} \qquad (17)$$

is a trapping rate to map the 3D setup onto a 1D diffusion problem. In this,

$$f(\sigma) = \frac{1 + A\sqrt{\sigma} - B\sigma^2}{(1-\sigma)^2} \qquad (18)$$

is a function that monotonically increases from 0 to infinity as $\sigma$, the fraction of the wall occupied by the circular PDs, increases from 0 to 1. $f(\sigma)$ is the result of a computer assisted boundary homogenization procedure with the values of $A$ and $B$ depending on the arrangement of trapping patches (*Berezhkovskii et al., 2006*). To maintain detailed balance, the corresponding trapping rate $\kappa_n$ must satisfy $A_w \kappa_w = A_n \kappa_n$, with $A_x$ the respective cross section areas of both tubes.

As PDs are very narrow, we must take into account that only part of the cross section surface inside the PD is available to a particle of size $\alpha$. Additionally, a subtle problem lies in the determination of $R_w$, as it is impossible to create a space filling packing with cylinders. To solve both issues, we rewrite *Equation 16* to explicitly contain cross section surfaces. We then replace $A_n$ with $\tilde{\tilde{A}}_n$ to accommodate hindrance effects and we replace $A_w$ by $1/\rho$. We also ajust PD length: $\tilde{l} = l + 2\alpha$ and $\tilde{L} = L - 2\alpha$. At the same time, we adjust $f(\sigma)$ to match a triangular distribution of the simplified PDs by using $A = 1.62$ and $B = 1.36$ (*Berezhkovskii et al., 2006*), which produces the hexagonal cytoplasmic column shape. This yields:

$$\tau_\parallel = \frac{1}{2D}\left[\tilde{L}^2 + \tilde{l}^2 + 2D\left(\frac{\tilde{l}}{\kappa_n} + \frac{\tilde{L}}{\kappa_w}\right) + \tilde{l}\tilde{L}\left(\tilde{\tilde{A}}_n\rho + \frac{1}{\tilde{\tilde{A}}_n\rho}\right)\right]. \qquad (19)$$

We similarly adjust $\kappa_w$:

$$\kappa_w = 4\rho D H_c(\alpha/R_n) R_n f(\rho\tilde{A}_n), \qquad (20)$$

where $H_c(\lambda)$ is the hindrance factor for cylindrical pores (see Materials and methods). In the same fashion, we also adjust $\kappa_n$.

We then invert the relation $\tau_\parallel = \frac{L^2}{2D} + \frac{L}{2P_{eff}}$, where we write $P_{eff}$ for the effective wall permeability (*Makhnovskii et al., 2009*), to obtain $P_{eff} = \frac{L}{2\tau_\parallel - L^2/D}$. With this, we can compute $f_{ih} = P_{eff}/(\rho\Pi(\alpha))$, where $\Pi(\alpha)$ is calculated using the same PD geometry. To validate the choice of boundary placement underlying the calculations above, we also calculated the MFPT over two PD passages, that is by shifting the reflecting boundary to the middle of one cell further. This resulted in a 4-fold increase of $\tau_\parallel$ and $L^2$ and hence in exactly the same $P_{eff}$.

To assess whether the desmotubule has a large impact on $f_{ih}$, we further adapt *Equation 19* by replacing $\tilde{\tilde{A}}_n$ by our desmotubule corrected $\tilde{\tilde{A}}_n$, except in $f(\sigma)$. Additionally, we multiply $f(\sigma)$ by $\xi = (\tilde{R}_n^2 - \tilde{R}_{dt}^2)/\tilde{R}_n^2$. Numerical calculations in a simple trapping setup confirm the validity of reducing $f(\sigma)$ proportional to the area occupied by the desmotubule whilst calculating $\sigma$ based on the outer radius alone (*Appendix 4—figure 1* and Appendix 4). This is in agreement with results for diffusion towards clusters of traps in 3D (*Makhnovskii et al., 2000*). By the same reasoning, we introduced a hindrance factor in $\kappa_w$. Finally, we adjust the hindrance factors to a slit geometry as before. This results in:

$$\tau_\parallel = \frac{1}{2D}\left[\tilde{L}^2 + \tilde{l}^2 + \frac{\tilde{L}/\rho + \tilde{l}\tilde{\tilde{A}}_n}{2R_n H(2\alpha/(R_n - R_{dt}))\xi f(\rho\tilde{A}_n)} + \tilde{l}\tilde{L}\left(\tilde{\tilde{A}}_n\rho + \frac{1}{\tilde{\tilde{A}}_n\rho}\right)\right]. \qquad (21)$$

To investigate the effect of different PD distributions, we used all relevant pairs of $A$ and $B$ in $f(\sigma)$ for different regular trap distributions as given in *Berezhkovskii et al. (2006)*. As $A_w$ is calculated implicitly from $1/\rho$, no other adjustments were necessary.

## Correction factor $f_{ih}$ for pit fields

For computing $f_{ih}$ in pit fields, we used a two step approach similar to computing $f_{ih}$ including DT as described above. A similar approach is also followed for the sub-nano channel model. In this calculation, a single pit field is modelled as a number of PDs on a triangular (or square) grid with a centre-to-centre distance $d$ between nearest neighbours. We then calculate the pit radius, $R_{pit}$ as the radius of the circle that fits the outer edges of the PD entrances. In the trivial case of one PD per 'pit', $R_{pit} = R_n$. For larger numbers of PDs per pit, see *Table 1*. For this calculation, individual PDs are modelled as straight cylindrical PDs with radius $R_n$. We calculate a $\tau_\parallel$ based on circular traps with radius $R_{pit}$ and a reduced efficiency based on the fraction of the pit that is occupied by the circular PDs. We accordingly adjust $\kappa_{w,pit}$ and $\tau_{\parallel,pit}$:

$$\kappa_{w,pit} = 4\rho D H_c(\alpha/R_{pit}) R_{pit} \xi f(\rho \pi \tilde{R}_{pit}^2), \tag{22}$$

where $p$ is the number of PDs per pit and $\xi = p\tilde{R}_n^2/\tilde{R}_{pit}^2$ is the fraction of available pit area that is occupied by available PD area, and

$$\tau_\parallel = \frac{1}{2D}\left[\tilde{L}^2 + \tilde{l}^2 + \frac{\tilde{L}/\rho + \tilde{l}p\tilde{\tilde{A}}_n}{2R_{pit}H(\alpha/R_{pit})\xi f(\rho\tilde{A}_{pit})} + \tilde{l}\tilde{L}\left(p\tilde{\tilde{A}}_n\rho + \frac{1}{p\tilde{\tilde{A}}_n\rho}\right)\right]. \tag{23}$$

In these equations, $\rho$ is the total PD density. In our graphs, we either keep $\rho$ constant while increasing $p$ to investigate the effect of clustering, resulting in a pit density $\rho_{pits}$ of $\rho/p$, or keep $\rho_{pits}$ constant to investigate the effect of (repeated) PD twinning. As a default, we used $d$ = 120 nm based on distances measured from pictures in *Faulkner et al. (2008)* of basal cell walls of *Nicotinia tabacum* leaf trichomes. To verify our calculations, we compared them with a single step calculation with large circles only, that is with radius $R_{pit}$ and density $\rho/p$. As results in 3D suggest that for strongly absorbing clusters, the outer radius and cluster density dominate the diffusion (survival time) process (*Makhnovskii et al., 2000*), this should produce a lower bound to $f_{ih}$. In terms of PDs, this regime applies if a particle that reaches a pit field also has a high probability of entering in it. Indeed, the values calculated with the two step method above were similar and somewhat larger than with the simple large patch method, showing that our computation method is reasonable.

Only a relatively small fraction of the pits is occupied by the PD entrances (5–10% when modelled as circles with $R_n$ = 14 nm and 3–7% with $R_n$ = 12 nm.). Consequently, this approach may become inaccurate when $R_{pit}$ gets too large. We indeed found instances where $f_{ih,pits}$ was larger than $f_{ih,singlePDs}$. In those cases, $R_{pit}$ was in the order of $d_{pit}/4$ or larger. We assume that in those cases, the clusters are so close, that the clustering has only minor impact on $f_{ih}$, and $f_{ih}$ is better estimated by the calculation for single PDs.

## Computing required densities or $\bar{\alpha}$ with default model

Numbers in *Table 2* are computed based on forward computation of $P(\alpha)$ given $\rho$, $\bar{\alpha}$, corresponding $R_n$ and other parameters with increments of 0.1 PD/µm² ($\rho$) or 0.01 nm ($\bar{\alpha}$ etc.) and linear interpolation between the two values that closest match the target $P(\alpha)$. This yields an error of less than 0.0001 µm/s on $P(\alpha)$. We use $\alpha$ = 0.5 nm for CF. The method for computing $P(CF)$ using the unobstructed sleeve (default) model is described throughout the main text. PDinsight, the python program used for computing all values in *Table 2*, *Appendix 5—table 1*, *Figure 8B* and *Table 1* is available as supporting material.

## Acknowledgements

The authors thank EM Bayer, W Nicolas, M Glavier and L Brocard from the Bordeaux Imaging Centre, Plant Imaging Platform, ((http://www.bic.u-bordeaux.fr), for providing the electron micrograph included in *Figure 1*. Also we thank Philip Kirk (Centre for Plant Sciences, Leeds) for critical review of the python program.

## Additional information

### Funding

| Funder | Grant reference number | Author |
|---|---|---|
| European Molecular Biology Organization | ASTF 105 - 2012 | Eva E Deinum |
| Nederlandse Organisatie voor Wetenschappelijk Onderzoek | | Bela M Mulder |
| Engineering and Physical Sciences Research Council | EF/M027740/1 | Yoselin Benitez-Alfonso |
| Leverhulme Trust | RPG-2016-13 | Yoselin Benitez-Alfonso |

The funders had no role in study design, data collection and interpretation, or the decision to submit the work for publication.

### Author contributions

Eva E Deinum, Conceptualization, Software, Formal analysis, Funding acquisition, Investigation, Visualization, Methodology, Writing—original draft, Writing—review and editing; Bela M Mulder, Supervision, Methodology, Writing—original draft, Writing—review and editing; Yoselin Benitez-Alfonso, Conceptualization, Supervision, Writing—original draft, Writing—review and editing

### Author ORCIDs

Eva E Deinum https://orcid.org/0000-0001-8564-200X
Bela M Mulder https://orcid.org/0000-0002-8620-5749
Yoselin Benitez-Alfonso https://orcid.org/0000-0001-9779-0413

### Decision letter and Author response

Decision letter https://doi.org/10.7554/eLife.49000.sa1
Author response https://doi.org/10.7554/eLife.49000.sa2

## Additional files

### Supplementary files

• Source code 1. Parameter files for PDinsight. PDinsight parameter files used in generating: table 1 + appendix 5 table 1 (tab1_pars.txt), table 2, numerical columns (e.g., fig7_pars.txt), figure 7 (fig7_pars.txt, fig7d_pars.txt), figure 8b (fig8b_pars.txt) and cross validation of figure 6 (fig6_pars.txt).

• Transparent reporting form

### Data availability

PDinsight can be downloaded from GitHub: https://github.com/eedeinum/PDinsight. Documentation on the use of PDinsight.py is included as an appendix to the manuscript with additional information at the head of the example parameter file. More extensive documentation is included with PDinsight on GitHub. PDinsight also has a citable DOI through Zenodo: 10.5281/zenodo.3536704. The PDinsight parameter files used for this manuscript are included as Source code 1.

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

## Appendix 1

# List of mathematical symbols

| | Default | Description |
|---|---|---|
| $\alpha$ | 0.5 nm for CF | Particle size |
| $\bar{\alpha}$ | | Maximum particle size that fits through a (model) PD |
| $A_c$ | | Cytosolic sleeve cross section area in the central region |
| $\tilde{A}_c$ | | Cytosolic sleeve cross section area in the central region adjusted for particle size (=steric hindrance only) |
| $\tilde{\tilde{A}}_c$ | | Cytosolic sleeve cross section area in the central region after cross sectional averaging (=full hindrance; $\tilde{\tilde{A}}_c = H(\lambda)A_c$) |
| $A_n$ | | Cytosolic sleeve cross section area in the neck region |
| $C$ | | Concentration |
| $\Delta C$ | | Concentration difference (over the PD channel) |
| $D$ | 162 $\mu m^2/s$ for CF | *Particle size dependent* diffusion constant |
| $d_1$ | 1 nm$^3$/s | Diffusion constant for particle of unit radius $\times 1$ nm (dummy value to illustrate scaling behaviour) |
| $f_{ih}$ | | Correction factor for inhomogeneous wall permeability ($0 \leq f_{ih} \leq 1$) |
| $H(\lambda)$ | | Hindrance factor calculated for a slit pore geometry |
| $H_c(\lambda)$ | | Hindrance factor calculated for a cylindrical pore geometry |
| $\lambda$ | | Relative particle size at the respective location (for straight PDs: $\lambda = \alpha/\bar{\alpha}$) |
| $l_n$ | 25 nm | Neck length |
| $\tilde{l}_n$ | | Effective neck length for a given particle size ($\tilde{l}_n = l_n + \alpha$) |
| $l$ | 100, 200, 500 nm | Total PD length |
| $L$ | 10 $\mu m$ | Cell length |
| $R_c$ | 17.5 nm | Central region radius |
| $R_{dt}$ | 8 nm | DT radius |
| $\tilde{R}_{dt}$ | | Particle size adjusted DT radius ($\tilde{R}_{dt} = R_{dt} + \alpha$) |
| $R_n$ | 12 nm | Neck radius |
| $R_x$ | | Central region or neck radius, depending on position |
| $R_{pit}$ | | Pit field radius: the radius of the smallest circle that circumscribes all PDs in the pit field |
| $\tilde{R}_x$ | | Particle size adjusted central region/neck radius ($\tilde{R}_x = R_x - \alpha$) |
| $\rho$ | | PD density |
| $p$ | 1 | Number of PDs per cluster ('pit field') |
| $P(\alpha)$ | | Symplasmic permeability (for particles of size $\alpha$) of the entire cell wall |
| $\Pi(\alpha)$ | | Symplasmic permeability (for particles of size $\alpha$) of a single PD per unit of cell wall surface, without correction factor $f_{ih}$ ($\Pi(\alpha) = \frac{P(\alpha)}{f_{ih}\rho}$). |
| $Q(\alpha)$ | | Molar flow rate through a single PD (for particles of size $\alpha$) |
| $Q_{rel}$ | | Molar flow rate relative to a reference situation (straight PD) |
| $\tilde{Q}_{rel}$ | | Rescaled $Q_{rel}$: $\tilde{Q}_{rel} = (Q_{rel} - 1)/(\frac{l}{2\tilde{l}_n} - 1) + 1$ |
| $\tau_{rel}$ | | Mean residence time (MRT) inside PD relative to a reference situation (straight PD) |
| $\tilde{\tau}_{rel}$ | | Rescaled $\tau_{rel}$: $\tilde{\tau}_{rel} = (\tau_{rel} - 1)l^2/(2\tilde{l}_n(l - 2\tilde{l}_n))$ |

*continued*

| | Default | Description |
|---|---|---|
| $\tau_\parallel$ | | Mean first passage time (MFPT) through a cytoplasmic column of length $L$ with a wall in the middle of it containing one central PD; Used for the calculation of $f_{ih}$ |

## Appendix 2

### Estimating systematic error due to homogeneous flux assumption

#### Molar flow rate

To estimate the error on the calculation of $Q(\alpha)$, we compared our analytical results (*Equation 2*) to numerical evaluations of the diffusion equations on a 2D cross section of the available surface of the model PD with neck, see *Appendix 2—figure 1*.

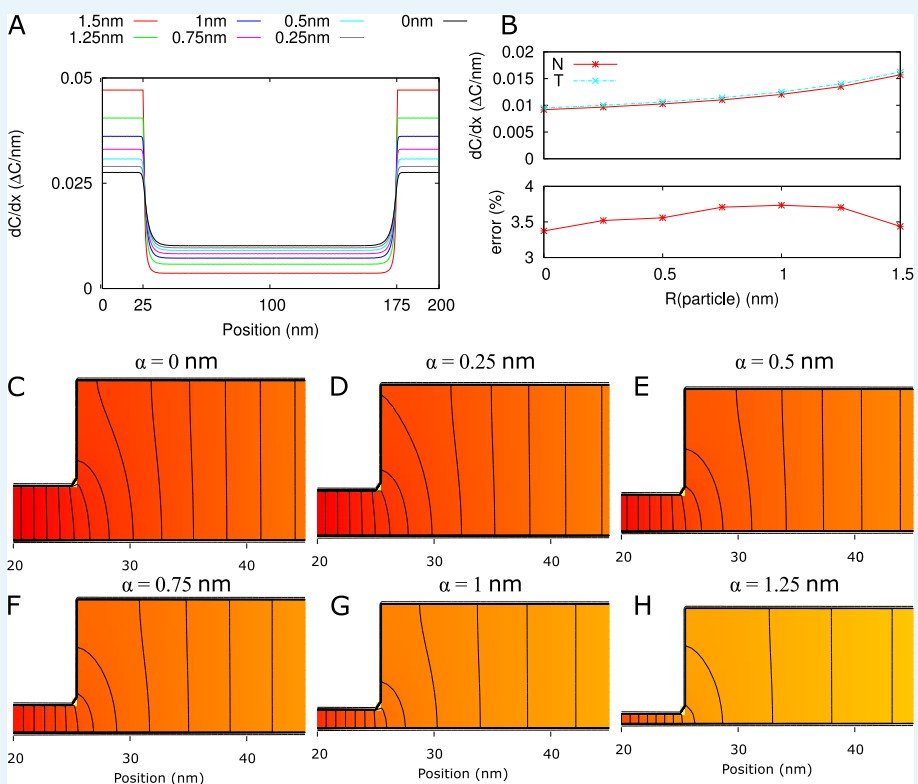

**Appendix 2—figure 1.** Error of homogeneous flux approximation (all 2D). (**A**) $\partial C/\partial x$ from numerical calculations (2D) along a straight line through the middle of the available neck region for different particle sizes. (**B**) Top: $\partial C/\partial x$ at neck entrance (proportional to the channel flux) from numerical calculations (N; solid red line with asterisks) and from 3-cylinder model with homogeneous flux assumption (T; dashed cyan line with crosses). The 3-cylinder model results in a consistent over estimation of < 4% (bottom). (**C-H**) Concentration heat maps for available part of the channel, focus on the neck/central region transition. The same color gradient is used for all six graphs. Black isolines are spaced at 1% of the total concentration difference over the channel. Parameters: $l$ = 200 nm, defaults.

#### MRT

In absence of a DT, we can compute $\tau_{\parallel,PD}$ analogously to $\tau_{\parallel}$. This yields:

$$\tau_{\parallel,PD} = \frac{1}{2D}\left(4\tilde{l}_n^2 + (l-2\tilde{l}_n)^2 + 2\tilde{l}_n(l-2\tilde{l}_n)\left(\frac{\tilde{\tilde{A}}_c}{\tilde{\tilde{A}}_n} + \frac{\tilde{\tilde{A}}_n}{\tilde{\tilde{A}}_c}\right) + \frac{\tilde{V}_c + 2\tilde{V}_n}{2R_nH(\lambda_n)f(\tilde{A}_n/\tilde{A}_c)}\right), \tag{24}$$

with $\tilde{V}_x$ the hindrance adjusted volume of central region or single neck region. The expression in brackets is identical to the one in *Equation 14*, except for the addition of the last term,

meaning that $\tau$ is an underestimation of the MFPT. Using $\tau_{\parallel,PD}$ to define an alternative $\tau_{rel}$ for channels without DT, $\tau_{\parallel,rel}$, suggests that $\tau_{rel}$ is an underestimate of at least approximately 7–9% for $R_c$ = 17.5 nm and $l$ = 100 nm (**Appendix 2—figure 2**). This factor saturates between 1.35 and 1.40 for all relevant particle sizes in the limit of unrealistically large $R_c$. For larger $l$, the factor has the same maximum values, but these are approached slower. Hence, the error is less for realistic $R_c$ (e.g. 4–6% for $l$ = 200 nm and 2–3% for $l$ = 500 nm.)

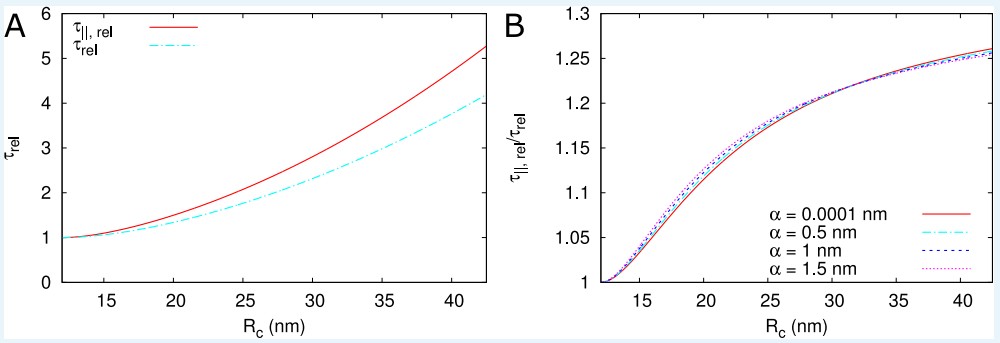

**Appendix 2—figure 2.** Comparison of $\tau_{rel}$ as used in the main text against a $\tau_{\parallel,PD}$ based calculation ($\tau_{\parallel,rel}$). $l$ = 100 nm, $R_n$ = 12nm, $R_{dt}$ = 8 nm, $l_n$ = 25 nm.

## Appendix 3

### Scaling behaviour of $Q_{rel}$ and $\tau_{rel}$

Rescaling of $Q_{rel}$ and $\tau_{rel}$ is a way to collapse our understanding of the processes into simpler curves. The local flux is inversely proportional to the available cross section, motivating the ratio $\tilde{\tilde{A}}_c/\tilde{\tilde{A}}_n$ of the dilated channel as a rescaling factor for the x-axis. Using the limit for $Q_{rel}$ it is possible to almost completely collapse the curves for different particle sizes for a single $l$, $l_n$ combination (*Appendix 3—figure 1B*). For rescaling of $\tau_{rel}$, we use its behaviour for large $R_c$:

$$\tau_{rel}(\alpha, R_c) \approx \frac{\tilde{\tilde{A}}_c(l - 2(\tilde{l}_n + \alpha))(2(\tilde{l}_n + \alpha))}{l^2 \tilde{\tilde{A}}_n}, \qquad (R_c \to \infty) \qquad (25)$$

$$\approx \frac{2\tilde{R}_c^2 \tilde{l}_n(l - 2\tilde{l}_n)}{l^2(\tilde{R}_n^2 - \tilde{R}_{dt}^2)}, \qquad (R_c \to \infty). \qquad (26)$$

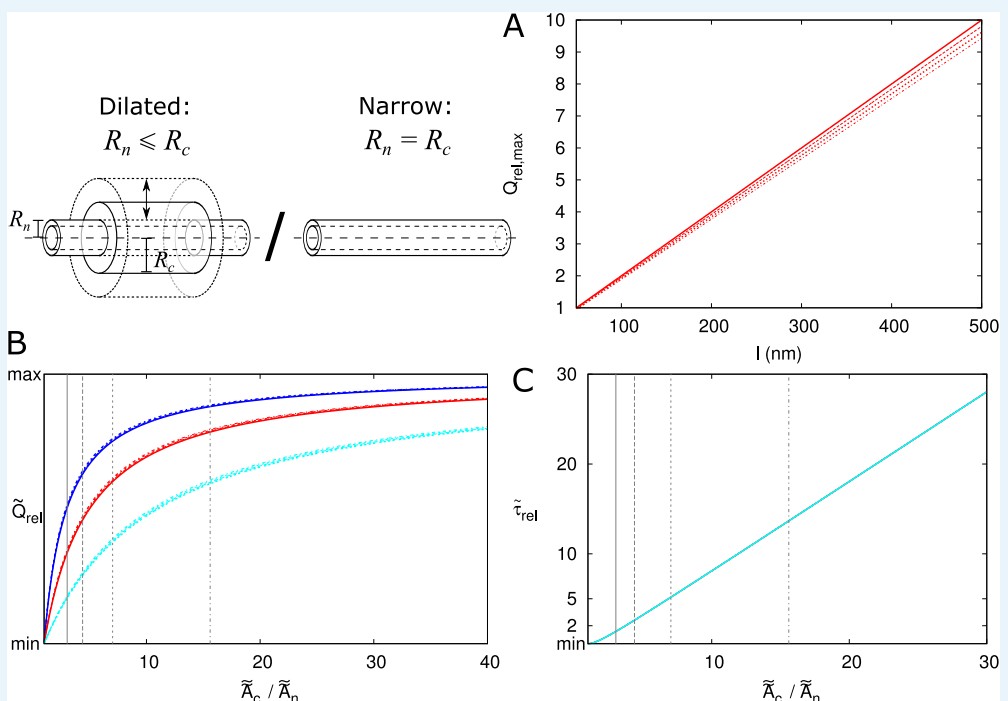

**Appendix 3—figure 1.** Scaling of $Q_{rel}$ and $\tau_{rel}$ through the PD. (**A**) Maximum possible increase of $Q$. Line type indicates particle size (solid: $\alpha \approx 0$, dashed: $\alpha = 0.5$ nm, sparse dashed: $\alpha = 1$ nm, dash-dotted: $\alpha = 1.5$ nm). (**B**) Curves for different $\alpha$ almost collapse with when plotted as function of $\tilde{A}_c/\tilde{A}_n$ and rescaled from the minimal value of 1 to $Q_{rel,max}$. The curves for different $l$ do not collapse (blue: $l = 100$ nm, red: $l = 200$ nm, cyan: $l = 500$ nm). (**C**) Curves for $\tau_{rel}$ collapse (for different $l$ and $\alpha$) with rescaling function $\tilde{\tau}_{re} = (\tau_{rel} - 1)l^2/(2\tilde{l}_n(l - 2\tilde{l}_n))$. Vertical lines in B,C correspond to $\tilde{\tilde{A}}_c/\tilde{\tilde{A}}_n$ for different particle sizes (line types as in A). Parameters: $l_n = 25$ nm, $R_n = 12$ nm, $R_{dt} = 8$ nm.

For large $R_c$, $\tau_{rel}$ becomes proportional to $\tilde{R}_c^2$. From this we derived a rescaling factor for $\tau_{rel}$ with $\tilde{f}_c$ the fraction of PD length occupied by the central region (adapted for particle size), that collapses the curves for $\tau_{rel}$ for all $\alpha$ and $l$ (*Appendix 3—figure 1C*). The rescaling factor for the x-axis, $\tilde{\tilde{A}}_c/\tilde{\tilde{A}}_n$, increases faster for larger particles. The reason is that $\tilde{\tilde{A}}_n$ decreases relatively faster with particle size than $\tilde{\tilde{A}}_c$, which becomes intuitively clear from *Figure 2C,D*. This difference explains why the curves for the largest particles are on top prior to rescaling (*Figure 4*). The $\tau_{rel}$-rescaling factor implies that the MRT increases fastest if the central region

occupies approximately half of the length of the channel. With our choice of a constant $l_n = 25$ nm, this occurs at a wall thickness of $l \approx 100$ nm.

## Appendix 4

### Numerical calculations for trapping rate of annular trap

Seen from the cytoplasmic bulk, the PD orifice has the shape of an annulus ('ring'). For our calculations above, however, we only have published trapping rates for circular traps. We tested two options for selecting an equivalent circular trap and trapping rate. For this, we numerically solved diffusion equations in a box with a single trap in the middle of one face and a source opposite to it (*Appendix 4—figure 1*). In the $x$ and $y$ direction, we used periodic boundary conditions reflecting a periodic array of traps. In the $z$ direction we fixed the concentration on side of the domain ('source plane') and used a radiating boundary condition with a mixed rate $\kappa(x, y) = k(x, y)D$ on the other side ('target plane'). We chose the trapping rate proportional to the diffusion constant, as the flux and molar flow rate through single PDs ($Q(\alpha)$) are proportional to $D$ (*Equation 2*). For PDs, the target plane contained the 'front view' of a single channel: an annulus with inner radius $R_{dt}$, outer radius $R_n$ and surface area $A_{ann}$. Within this annulus $k(x, y)$ was set to unity ($k(x, y)|_{PD} = 1/\mu m$) and 0 outside. For the corresponding homogeneous target plane $\kappa_{hom} = \frac{A_{ann}}{A_{total}}/\mu m$. At the grid level, the pixels at a boundary of the annulus had $\kappa$ proportional to the fraction of their surface falling inside the annulus. The reference flux was computed analytically, exploiting that within each plane at a given distance $z$ from the target plane, the concentration is the same everywhere. This allows for a trivial mapping to a 1D system. These 3D numerical calculations were performed using the Douglas method for 3D alternating direction implicit diffusion (*Douglas and Gunn, 1964*). To save computation time we used the analytically calculated reference profile as an initial condition for all calculations. We found that the annular patches gave the same result as circles with radius $R_n$ and $k(x, y)|_{circ} = \frac{A_{ann}}{A_{circ}}/\mu m$, that is that the outer radius of the patch was most important (*Appendix 4—figure 1*), In line with results for diffusion towards clusters of traps in 3D (*Makhnovskii et al., 2000*). These calculations support the choice of trapping rate in the calculation of $f_{ih}$ including DT.

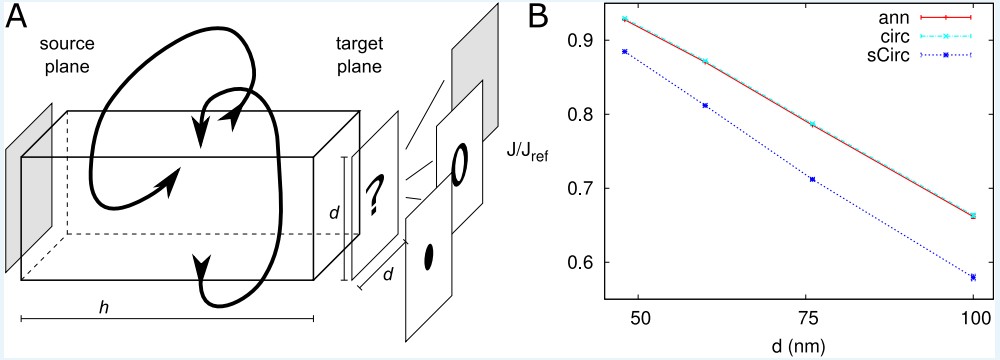

**Appendix 4—figure 1.** Correction factor for annular trap shape. Comparison of discrete and homogeneous permeability. (**A**) Setup. A homogeneous source is located at a distance $h$ from a trapping plane with either a single trap or homogeneous absorbency. The periodic boundary conditions in the other directions make that the traps are effectively spaced on a square grid with distance $d$ between traps. (**B**) Relative average net flux in direction of trap (component along the box) for PDs with DT ('ann': red) and circular channels with the same apparent surface ('sCirc': blue) and same outer radius, but trapping rate decreased according to surface ratios ('circ': cyan), $h$ = 300 nm. For both, the flux is compared with a homogeneous trap with rate $\kappa = \frac{A_{ann}}{A_{total}}D/\mu m$. Error bars (nearly invisible) represent the quality of the numerical estimate by minimum and maximum possible values.

## Appendix 5

### Computing required densities or $\bar{\alpha}$ with sub-nano model

Required densities or $\bar{\alpha}$ for the sub-nano model are computed with PDinsight similar to the default model. Additionally, we use the following approximations. For computing $\rho_{sn}$ based on a given $\bar{\alpha}$, we compute a density multiplier based on $n_c$ (**Equation 7**) and **Figure 5**. This implicitly assumes that $f_{ih}$ is not affected by the different PD density. For example, for $\bar{\alpha}$= 2.5 nm, a 16.8 (**Equation 7**) × 1.63 (**Figure 5**) / 9 (channels per PD; **Olesen, 1979**; **Liesche and Schulz, 2013**) = 3.04 times as high density would be sufficient. These numbers are the same for straight PDs of any length. For calculating $\rho_{sn,neck}$, we assume that the sub-nano channel structure only occurs in the neck region, with an unobstructed sleeve with $R_c = R_{dt} + 2\bar{\alpha}$ in the central part. Using a homogeneous flux assumption around the transition between both parts, the factors $x$ reduce to $x(2\tilde{l} + (1 - 2\tilde{l})/x)/l$. Similarly, $\rho_{gate}$ is computed by assuming a 1 nm thick sub-nano channel structure at both ends of the PD.

As $f_{ih}$ is affected by $R_n$ and $R_n$ values get quite large in our calculations with $\rho$ fixed, we follow a different approach for computing $\bar{\alpha}_{sn}$ and $R_{n,sn}$ based on a given $\rho$. We use forward calculations based on nine cylindrical channels in a PD, with the trapping rate $\kappa_w$ adjusted with an outer radius $R'_n$ that would fit all nine channels surrounding the DT.

$$\kappa_{w9} = 4\rho D H_c(\alpha/R'_n)R'_n \xi f(\rho \pi \tilde{R}'^2_n), \tag{27}$$

where $\tilde{A}_n$ is the surface area per cylindrical channel and $\xi = \frac{9(\bar{\alpha} - \alpha)^2}{\tilde{R}^2_n}$ is the fraction of the enveloping circle that is occupied by the nine channels. For sufficiently small $\bar{\alpha}$, the nine circular channels and minimal protein spacers (at least 1 nm wide) all fit while touching the DT. In that case: $R'_n = R_n = R_{dt} + 2\bar{\alpha}$. With $R_{dt}$= 8 nm, this is possible up to $\bar{\alpha} \approx$ 3.4 nm. For larger $\bar{\alpha}$, the spacer requirement determines the outer radius of the composite of 9 channels and $R'_n = \bar{\alpha} + \frac{\bar{\alpha} + s/2}{\sin(\pi/9)}$, where $s$=1 nm is the spacer width. $R_{dt}$ does not occur in this equation, because the cylindrical channels can no longer (all) touch the DT.

**Appendix 5—table 1.** Parameter requirements for reproducing measured $P(CF)$ values based on the default unobstructed sleeve model (also in *Table 2*) and the sub-nano channel model. A: Required density ($\rho$) given $\bar{\alpha}$ and corresponding neck radius ($R_n$). B: Required $\bar{\alpha}$ and corresponding $R_n$ given $\rho$. For P (CF) = 25$\mu$m/s, also values for a 2x, 3x and 4x increase of $\rho$ are computed. This is done both for a uniform increase of the density ($p = 1$) and for (repeated) twinning ($p>1$) from a uniform starting density (indicated in bold). A, B: The + sign at $R_n$ indicates that the stated $R_n$ is too narrow to fit nine sub-nano channels that touch a DT with $R_{dt} = 8$ nm. Models used for calculating required densities: $\rho$, $R_n$: default unobstruced sleeve model, $\rho_{sn}$, $R_{n,sn}$: sub-nano channel model, $\rho_{sn,neck}$: sub-nano channel model restricted to the neck regions ($l_n$= 25 nm), and $\rho_{gate}$: 1 nm thick structures at both PD entrances locally similar to the sub-nano channel model.

**A**

| $P(CF)$ ($\mu$m/s) | $\bar{\alpha}$ (nm) | $R_n$ (nm) | $\rho$ (PD/$\mu$m$^2$) | $\rho_{sn}$ | $\rho_{sn,neck}$ | $\rho_{gate}$ |
|---|---|---|---|---|---|---|
| 6 | 2.0 | 12 | 33.5 | 139.9 | 87.8 | 36.7 |
| | 2.5 | 13 | 22.7 | 69.1 | 46.4 | 24.1 |
| | 3.0 | 14 | 16.8 | 40.9 | 29.1 | 17.5 |
| | 3.4 | 14.8 | 13.8 | 29.1 | 21.6 | 14.2 |
| | 3.5 | 15+ | 13.2 | 27.0 | 20.2 | 13.6 |
| | 4.0 | 16+ | 10.7 | 19.2 | 15.0 | 10.9 |
| 8.5 | 2.0 | 12 | 47.2 | 197.1 | 123.7 | 51.7 |
| | 2.5 | 13 | 32.0 | 97.4 | 65.4 | 34.0 |
| | 3.0 | 14 | 23.7 | 57.7 | 41.0 | 24.7 |
| | 3.4 | 14.8 | 19.4 | 41.1 | 30.5 | 20.1 |
| | 3.5 | 15+ | 18.5 | 38.1 | 28.5 | 19.1 |
| | 4.0 | 16+ | 15.0 | 27.0 | 21.1 | 15.4 |

**B**

| $P(CF)$ ($\mu$m/s) | $\rho$ | $\bar{\alpha}$ | $R_n$ (nm) | $\bar{\alpha}_{sn}$* | $R_{n,sn}$(nm) no spacers | $R_{n,sn}$ 1 nm spacers |
|---|---|---|---|---|---|---|
| 6 | 10 | 4.2 | 16.3 | 5.2 | 18.4+ | 21.8 |
| | 13 | 3.5 | 15.1 | 4.6 | 17.3+ | 19.6 |
| 8.5 | 10 | 5.2 | 18.4 | 6.0 | 20.1+ | 25.2 |
| | 13 | 4.4 | 16.8 | 5.4 | 18.7+ | 22.5 |
| 1 | 10 | 1.5 | 11.0 | 2.6 | 13.2 | 13.2 |
| | 13 | 1.3 | 10.6 | 2.4 | 12.7 | 12.7 |

**C**

| $P(CF)$ ($\mu$m/s) | $\rho$ | $p$ | $\bar{\alpha}$ | $R_n$ (nm) | $\bar{\alpha}_{sn}$* | $R_{n,sn}$(nm) no spacers | $R_{n,sn}$ 1 nm spacers |
|---|---|---|---|---|---|---|---|
| 25 | **10** | **1** | **10.5** | **28.9** | **10.0** | **28.0+** | **40.7** |
| | 20 | 1 | 6.6 | 21.2 | 7.1 | 22.2+ | 29.4 |
| | | 2 | 7.2 | 22.5 | 7.5 | 23.1+ | 31.1 |
| | 30 | 1 | 5.1 | 18.1 | 5.9 | 19.8+ | 24.6 |
| | | 3 | 5.6 | 19.2 | 6.3 | 20.6+ | 26.3 |
| | 40 | 1 | 4.2 | 16.4 | 5.2 | 18.4+ | 21.8 |
| | | 4 | 4.6 | 17.2 | 5.5 | 19.0+ | 23.1 |

*Appendix 5—table 1 continued on next page*

| 13 | 1 | 8.8 | 25.6 | 8.8 | 25.5+ | 35.9 |
|---|---|---|---|---|---|---|
| 26 | 1 | 5.6 | 19.1 | 6.3 | 20.6+ | 26.2 |
| | 2 | 6.0 | 20.0 | 6.6 | 21.2+ | 27.3 |
| 39 | 1 | 4.3 | 16.6 | 5.2 | 18.5+ | 22.1 |
| | 3 | 4.6 | 17.2 | 5.5 | 19.0+ | 23.1 |
| 52 | 1 | 3.6 | 15.1 | 4.6 | 17.3+ | 19.6 |
| | 4 | 3.8 | 15.5 | 4.8 | 17.6+ | 20.4 |

*: $\bar{\alpha}_{sn}$ is calculated using $R_n$ that allows for 1 nm spacers. C: $p$ is the number of PDs per pit. This table was generated using PDinsight.

**Appendix 6**

## PDinsight

PDinsight is written in python 3. If available, it uses the numpy module, but does not strictly depend upon that. The program has different modes for computing the parameter requirements for a given $P(\alpha)$ and related quantities. The different modes and the relevant parameters are controlled from a parameter file (default: parameters.txt). A graphical user interface (GUI) is provided to help the user create parameter files and run PDinsight. The GUI is written using TKinter, which is included in standard installation of python.

For electron microscopists, who typically have access to many ultrastructural parameters, but often do not know $P(\alpha)$, the mode computeVals will be useful. This computes the expected $P(\alpha)$ values when taking all parameters at face value. Comparison with the sub-nano channel model is possible. In principle, all model parameters must be defined, but missing parameters may be explored using lists of possible values or left at a default (e.g., cell length $L$ and a triangular distribution of PDs), as these have little influence on $P(\alpha)$. If estimates of PD density and distribution are missing, the mode computeUnitVals, which computes $\Pi(\alpha)$, could be useful. In this mode, comparison with the sub-nano channel is impossible.

In tissue level experiments, $P(\alpha)$ is typically measured, but not all ultrastructural parameters will be known. It is likely that PD density $\rho$ and radius ($R_n$, assuming straight channels) are poorly known. In this case, mode computeRnDensityGraph will be useful, or a combination of modes computeDens and computeAperture and a number of guesses for $R_n/\bar{\alpha}$ or $\rho$, respectively. Additional poorly known parameters can be explored as suggested above. If uncertain, it is strongly advised to explore PD length $l$ ($\approx$ cell wall thickness). For thick cell walls ($l \geq 200$ nm), it may be worth exploring the effects of increased central radius ($R_c > R_n$). This is currently only possible in modes computeVals and computeUnitVals.

## Major modes

The core of PDinsight is computing effective permeabilities ($P(\alpha)$) for symplasmic transport based on all model parameters mentioned in the manuscript. This is in mode computeVals. The same computations are used in other modes, which compute the requirements for obtaining a given target value (or set of values) of $P(\alpha)$. In mode computeDens, required densities ($\rho$) are computed for given values of maximum particle size $\bar{\alpha}$ (and other parameters). In mode computeAperture, required apertures, given as $\bar{\alpha}$ as well as neck radius $R_n$, are computed for given values of $\rho$ (and other parameters). In mode computeRnDensityGraph, $R_n, \rho$ curves are computed that together yield a target $P(\alpha)$. The corresponding values of $\bar{\alpha}$ are also reported. These curves can be visualized using any plotting program. Mode sensitivityAnalysis computes so-called elasticities (normalized partial derivatives) around a given set (or sets) of parameters. These elasticities tell how sensitive calculated values of $P(\alpha)$, $\Pi(\alpha)$ and constituents like $f_{ih}$ are on the parameters involved.

## Auxillary modes

In computing $P(\alpha)$, correction factor $f_{ih}$ is automatically included. For specific cases such as modelling studies, however, it may be useful to calculate $f_{ih}$ separately. For this purpose, several modes exist for exploring inhomogeneity factor $f_{ih}$: computeFih_subNano (function of $\bar{\alpha}$; also output values for sub-nano model), computeFih_pitField_dens (function of $\rho$), computeFih_pitField_xMax (function of $\bar{\alpha}$) and computeTwinning (function of $\rho_{pits}$, cluster density).

By default, computations are performed for the unobstructed sleeve model. Most computations can also be performed for the sub-nano channel model (*Liesche and Schulz, 2013*; *Comtet et al., 2017*). Using switch compSubNano, values for the sub-nano model are also computed.

## Graphical user interface

: The GUI to PDinsight is written to facilitate the creation of parameter files and also has a button to run PDinsight directly based on the parameters displayed. In contrast to the generic parameter file, the GUI only shows fields for parameters that are actually used in a specific run mode. The first step of using the gui is to select the run mode (choose from: `computeVals`, `computeUnitVals`, `computeDens`, `computeAperture`, `computeRnDensityGraph` and `sensitivityAnalysis`), followed by 'load default parameters'. This overwrites any user input from previous modes. All required parameters are shown as text entry fields, with radio buttons for the relevant options. Basic validation occurs on the fly (valid input type, etc). When done, click 'Run PDinsight' to write a parameter file and run the program or 'Export parameter file' to write a parameter file only. Note that switching modes requires clicking 'load default parameters', that is a fresh start.

Additional information can be obtained by clicking the 'Info' button and, for certain parameters, clicking on the parameter name. If additional information is available, the mouse cursor changes into a question mark.

## Command line usage

(linux): python PDinsight_vXX.py PARAMETERFILE.

(windows): first open a command prompt window (e.g., by searching for 'cmd' in the search bar) and go to the directory where PDinsight is located. Then type: py PDinsight_vXX.py PARAMETERFILE

Using the GUI from the command line (linux): python PDinsightGUI_vYY.py or (windows): py PDinsightGUI_vYY.py The correct version of PDinsight_vXX.py should also be available in the current directory.

In the above, XX and YY should be replaced by the respective version numbers.

By default, all outputs are tab separated files (.tsv). The other option is comma separated (.csv), which can be set through the parameter outputType.

The latest version of PDinsight + documentation can be downloaded from https://github.com/eedeinum/PDinsight. DOI: 10.5281/zenodo.3536704

