## [Decision Letter]

**Acceptance summary:**

Plasmodesmata (PD) are highly regulated channel-like structures between plant cells that enable movement of diverse molecules including metabolites, proteins and viruses. Long standing questions include how the PD is structured and how this structure enables selectivity that changes during plant growth and upon stress.

In this manuscript, the authors present a mechanistic model to investigate the relation between the geometry of PD, molecular diffusion rate across PD, and molecule residence time in the PD. Key achievements are the scaling of single PD models to whole cell-cell interfaces, providing valuable insight on the design principles of PD and their distributions, as well as providing a tool to predict flux based on structural parameters. This is especially important for internal plant tissues that are not accessible for many experimental approaches. The major novelty of the model is that it is based on a realistic description of PD geometry and mostly relies on measurable parameters. It is a valuable companion to the direct measurements of symplasmic fluxes using molecular reporters and new microscopy technologies allow the fast acquisition of datasets about PD geometry, but that cannot be exploited to their full potential without modelling tools. The release of an open source version of their model constitutes a substantial service to the plant biology community.

**Decision letter after peer review:**

Thank you for submitting your article "From plasmodesma geometry to effective symplastic permeability through biophysical modelling" for consideration by *eLife*. Your article has been reviewed by three peer reviewers, and the evaluation has been overseen by a Reviewing Editor and Christian Hardtke as the Senior Editor. The following individuals involved in review of your submission have agreed to reveal their identity: Johannes Liesche (Reviewer #1); Michael Knoblauch (Reviewer #2); Valentin Couvreur (Reviewer #3).

The reviewers have discussed the reviews with one another and the Reviewing Editor has drafted this decision to help you prepare a revised submission.

Through discussions among the reviewers it was clarified that further experiments would improve the manuscript, but were not strictly necessary for the work to be a strong advance for the field, so long as a number of textual changes were made give a clear context for this work in light of previous studies. For this reason, I am compiling the extensive notes from the three reviewers as a list below. Please consider these previous data when writing a revision and a response.

Overall, it will be essential to demonstrate that the model has broad applicability, but to recognize that it was not yet tested in many situations.

1) Since the model was only applied to a single interface, and reviewer one seems to think the data basis seems to be handled loosely, please see comments from reviewer 1 to address this issue.

2) A more careful evaluation of the current literature, including the discussion of previous models by various authors, is needed.

3) The authors should consider if their model could be even more ambitious, for example by including evaluation of neck length and non-straight shapes.

Reviewer 1:

Introduction paragraph four: The paper by Liesche and Schulz, 2013, is not presented here accurately. This paper compares three different models of PD anatomy. One of them is a cytosolic sleeve model that shares quite some similarity with the model proposed here. The sub-nano channel model was only developed for a very specific question: whether PD can be constricted enough to enable the filter effect that has been ascribed to them as part of the polymer-trap mechanism for phloem loading. According to the hypothesis, sucrose should be able to diffuse through the PD at the bundle-sheath-to-phloem interface, but the slightly larger raffinose should not. It was the aim of that paper to test what kind of PD architecture could be compatible with this mechanism.

Introduction paragraph five: There is no principle limit to using fluorophores of different sizes. Terry and Robards, 1987, actually used tracers of all sizes. Only because fluorescein, especially as carboxyfluorescein diacetate (CFDA) it is widely used, because it crosses membranes, allowing for non-invasive observations.

Modern tracer-based approaches using confocal microscopy can yield very similar results. Compare Rutschow et al., 2011 and Liesche and Schulz, 2013. See also Liesche and Schulz (2012, Plant Physiology), who compared permeabilities across plant species and different cell-cell interfaces.

Photoactivation and photobleaching approaches are not time consuming. Quite the opposite. Application of tracer, sample preparation and imaging can easily done within half an hour.

Introduction paragraph six: I suggest formulating with greater care. Photoactivation and photobleaching approaches as well as GFP transport studies have been very valuable tools for assessing PD function.

The incompatibility here could have very different reasons. A better citation would be Liesche et al., 2019, which actually tries to compare structure-based modeling with functional data. As mentioned above, the Liesche and Schulz, 2013, paper had a different objective.

General comment to the Introduction: I advise a more nuanced view at the current state of the field and approaches carried out in the past. Also, I recommend using the Liesche et al., 2019 paper in Plant Physiology as starting point because it actually is the first systematic comparison of PD structure and function. Moreover, this publication clearly demonstrates the need for better models.

Figure 1: The effective symplasmic permeability in a file of cells also depends on cell size and properties of the cytosol.

Final paragraph of the Introduction: Python is a start, but how about making a standalone app? Or an add-in/macro for Excel? Something that any molecular biologist (and especially students) can use! Last year I got quite a bit of experimental data on phloem transport and I wanted to check it against theoretical models that have been described as easy to use. My basic Matlab skills were not enough to make it work. In my opinion, if you are serious about getting a lot of scientists to try the model, you have to provide a truly easy-to-use solution.

Results subsection “Outline of the model”: Many of the equations used in this model were applied to calculate PD transport for the first time by Liesche and Schulz, 2013. Maybe that should be acknowledged somewhere in this paragraph or the Introduction.

Figure 3: What is the reasoning behind plot B? How can I see from the plot that D always has a big influence (which, of course, is not unexpected)? All values of permeability are very low.

Results paragraph five: density jump is not a term often used in PD literature. You could consider using concentration potential.

Subsection ““Necked” PDs increase molecular flux in thicker cell walls”: I find this heading slightly misleading. Shouldn't it be "dilated PDs increase molecular flux"? Because *R_n_* is the same in both cases, meaning that the neck doesn't change. Or am I just misled by the drawing in Figure 4?

First paragraph of subsection ““Necked” PDs increase molecular flux in thicker cell walls”: Maybe write "electron microscopy" instead of "evidence".

Figure 4: Again, "dilated" seems more appropriate than "necked". Panel C has a different y axis scale from all the other panels. The legend in panel C is confusing as it is missing an α. I recommend either legends in all panels or description in the figure caption.

Figure 4 and accompanying text: Is *R_n_* = 12nm in the 'necked' PD and 17.5 in case of narrow PD? This should be formulated clearly, both in the figure and the text. The drawing above panel A gives the impression that they have the same *R_n_*.

Paragraph three of the same section: In this case the length of the neck region is still 1/3rd of the total PD length, right? It might be worthwhile to also check the effect of variation in the length of the necked region. Ideally, this should also be one parameter of the python tool, because it can be estimated from TEM images.

Subsection “The desmotubule increases PD transport and changes the dependence on particle size”: I also find this heading misleading. My first intuition would be to compare a PD with DT to a PD without DT.

In the same subsection what exactly does the assumption on selection towards size-selectivity mean? That a plant would make the PD as narrow as possible, depending on the molecule that is supposed to pass? This assumption might be valid for relatively big molecules, but probably not for small ones (below 1-2nm). Except for the very special PDs in the polymer trapping species mentioned earlier. I recommend discussing this assumption in more detail.

Where exactly does the "see methods" refer to? I don't find a section on DT-related calculations in the methods.

Paragraph two of subsection “Clustering of PDs in pit 1elds reduces effective symplastic permeability”: Wasn't *R_n_* = 12 nm in Figure 4?

In paragraph three of the same section: Rpit is not in the list of mathematical symbols. Moreover, it is not clear why the radius of a pit field should be comparable to the radius of individual PD, since the number and size of PD within a pit field can vary.

In the final paragraph of the same section: How should this be measurable? As a correlation of clustering with wall thickness?

Figure 7: In panel A, is the distance between PD or the radius of the pit field constant?

Paragraph three of subsection “Application of the model to compute effective permeability for fluorescein derivatives”: With a density of 5 (as specified by Zhu et al., see comments below), it seems like the value for *R_n_* should be 22 nm.

Paragraph four: Why is a density of 10 μm^2^ used here instead of the value from Zhu et al. (5.4 μm^2^)?

Also didn't a previous paragraph show that central dilation ("necking") is especially effective for long PDs? How does that fit with the results presented here?

Table 1: PD density seems to start at 10 μm^2^. This value is higher than the 5.4 μm^2^ specified by Zhu et al. (Table 2). Their value should be considered a max value as the area that was analyzed by Rutschow et al. is reaching into the mature root zone already. Mature zone has much lower density values (0.62 μm^2^, Table 3 in Zhu et al.).

In paragraph five the model seems incompatible with the low-concentration H_2_O_2_ data. At a density of 5.4 μm^2^, *R_n_* would need to increase to over 40 nm according to Figure 8B. This seems unrealistic. An increase in PD number can also not be expected because of the short treatment period of 2h.

“We also compared the results obtained with our model and the sub-nano channel model reported before (Liesche and Schulz, 2013).”: This makes very little sense as this model was put forward only for a very specific case, as mentioned above. It would be better to compare with the cytosolic slit model that was also described in that paper.

Discussion paragraph two: Again, this model should not be compared here or it should be clearly stated that it was not developed to be applied in this way. Instead the authors should compare their model to the cytosolic sleeve model of Liesche and Schulz (2013) or the 'pure diffusion' model by Doelger et al., 2014 or the model for 'diffusion through simple PD' included in Ross-Elliott et al., 2017.

“For example, sucrose moves symplastically from bundle sheet cells (BSC) to intermediary cells (IC), where it is polymerized into the larger oligomers raffinose and stachyose, that do not diffuse back in detectable amounts”: It should be mentioned that this only happens in certain species (active symplasmic phloem loaders) with the Cucurbits as the most prominent example.

Discussion paragraph four: The Liesche et al., 2019, paper does not analyze correlation of PD length with permeability at the BSC-phloem interface of active symplasmic loaders. It analyzes correlations of PD anatomical parameters (including length) with permeability across species. I don't understand why a correlation of PD length with permeability would be expected, as the study generally finds permeability to depend to a much higher degree on PD diameter. Please also note that it is very unlikely that a bulk flow was overlooked in that study as the measurements were performed on detached leaves. Indeed, this has been tested by Liesche and Schulz, 2012 who compared full photosynthesising leaf and cut-out tissues and found no difference in PD permeability in tobacco.

Discussion paragraph six: please include 'at the root unloading zone'. Funnel PDs are not found at other phloem interfaces.

“Applying our model for diffusion as a sole driver of symplastic transport can indeed explain experimentally observed measurements of effective symplastic permeability for CF, but only with somewhat wider PDs/neck regions or several fold higher PD densities than usually measured by EM.”: This statement seems to contradict itself. In addition, I highly recommend application of the model to additional interfaces. Liesche et al., 2019, for example, found a model that matches observations of permeability for the bundle-sheath-phloem interface, but not for the bundle sheath-mesophyll interface. In the present case, it should be noted that PD density and dimensions were determined for sand-grown plants (Zhu et al., 1998), whereas effective diffusivity was measured for plate-grown plants (Rutschow et al., 2011). I don't know how growth conditions affect PDs, but it has been previously shown that there are big differences, for example in gene expression, between soil- and plate-grown plants. I mention this just to illustrate the need for additional tests.

“Our model can also explain the effect on permeability after treatment with high and low concentrations of H_2_O_2_ in Rutschow et al., 2011.”: Please reconsider this statement regarding low H_2_O_2_ in light of comments above.

I am very skeptical about the conclusion. Stress perception, signaling, and two rounds of twinning, all within 2h seem unrealistic. I also didn't find evidence for such a rapid multiplication of PD numbers in the two citations provided here. Moreover, 30 PD μm^2^ would be an extremely high number, that, to my knowledge, has never been observed at a "regular" cell-cell interface.

Discussion paragraph seven, “Despite these deviations, comparing our model to the sub-nano channel model, we found that the latter requires roughly twice as high PD densities to produce the same permeability values *P (CF)*” and following sentences: Again, this is not an appropriate comparison. Please compare to other models listed in a comment above.

---

## [Author Response]

Through discussions among the reviewers it was clarified that further experiments would improve the manuscript, but were not strictly necessary for the work to be a strong advance for the field, so long as a number of textual changes were made give a clear context for this work in light of previous studies. For this reason, I am compiling the extensive notes from the three reviewers as a list below. Please consider these previous data when writing a revision and a response.Overall, it will be essential to demonstrate that the model has broad applicability, but to recognize that it was not yet tested in many situations.

We now explicitly discuss the need for additional testing and its expected benefits in the manuscript. To illustrate broad applicability, we have analyzed three additional experiments where pure diffusive transport was expected (an epidermis only experiment in Rutshow et al., 2011, from which we already analyzed the other experiments; additional data obtained from Johannes Liesche (reviewer 1), intended for a follow up manuscript analyzing even more data. See responses to reviewer 1 and Table 1 for more details. As argued in the introduction, pure diffusive symplasmic transport is likely to be the relevant form of symplasmic transport in many developmentally relevant processes/conditions. Additionally, we have added a short section to the Discussion describing what would be needed to include advection/flow while conserving the unique features of our model. Finally, to allow more researchers to join in this broader application effort, we have put considerable effort in building a graphical user interface to the python program, which, according to testers, was makes it much easier to use.

1) Since the model was only applied to a single interface, and reviewer one seems to think the data basis seems to be handled loosely, please see comments from reviewer 1 to address this issue.

We assume that the suggestion that “the data basis seems to be handled loosely” is mostly based on the comments by reviewer 1. As described there, we disagree on the remark that we should have used a 2-fold lower PD density. The density of ~5PD μm^2^ applies to the epidermis. About 2 times higher densities are reported for more interior layers, which comprise the bulk of the tissue in the experiment we analyze. For consistency, we now also analyzed the second measurement from the Rutschow paper, which is limited to the epidermis and where lower permeabilities are reported from the experiment. In our analysis, the different permeabilities reported in Rutschow et al., 2011 and the different densities for different interfaces reported in Zhu et al., 1999 are all consistent.

To test for broader applicability, we have also applied the model (with slight addition of functionality) to two data sets originally acquired in the context of Liesche et al., 2019: mesophyll-mesophyll interfaces in poplar and Cucurbita max. As stated in the general response, we have included the results of this analysis in a separate document for internal use in the reviewing process, but prefer not to include them in the current manuscript but make them part of a larger analysis / collaborative follow up project, 1) to avoid a potential conflict of interests and 2) because their inclusion would not cancel the suggestion that further testing remains highly valuable (and is an important reason for building a community tool).

2) A more careful evaluation of the current literature, including the discussion of previous models by various authors, is needed.

We have incorporated a discussion of previous models by various authors in the Introduction. See responses to reviewer 1 and general comment on the Introduction for details. We also spend more words in the discussion comparing our model to various other models using a similar PD architecture.

3) The authors should consider if their model could be even more ambitious, for example by including evaluation of neck length and non-straight shapes.

We have included a figure on the impact of varying neck length between 15 – 50 nm (Figure 4—figure supplement 2). Variation of neck length (and different neck lengths/diameters on either side of the PD) has now been included in PDinsight in the process of analyzing the additional data. The suggestion of non-straight shapes is certainly interesting and envisioned as a future extension of PDinsight, but careful implementation of this feature is not feasible within the two months given for resubmission and, moreover, would add yet another topic to an already dense manuscript.

Reviewer 1:Introduction paragraph four: The paper by Liesche and Schulz, 2013, is not presented here accurately. This paper compares three different models of PD anatomy. One of them is a cytosolic sleeve model that shares quite some similarity with the model proposed here. The sub-nano channel model was only developed for a very specific question: whether PD can be constricted enough to enable the filter effect that has been ascribed to them as part of the polymer-trap mechanism for phloem loading. According to the hypothesis, sucrose should be able to diffuse through the PD at the bundle-sheath-to-phloem interface, but the slightly larger raffinose should not. It was the aim of that paper to test what kind of PD architecture could be compatible with this mechanism.

We originally cited Liesche and Schulz, 2013 for this model architecture as the first to model this architecture. To avoid confusion, we have emphasized the specific context used in Liesche and Schultz, 2013, which is the same context as in Comtet et al., 2017, now also cited for using the sub-nano channel architecture.

The sub-nano architecture is now part of a section that describes the generally used architectures, i.e., a cytosolic slit without further localized obstructions and the sub-nano channel structure. Liesche and Schultz, 2013 is cited in both categories.

Introduction paragraph five: There is no principle limit to using fluorophores of different sizes. Terry and Robards, 1987, actually used tracers of all sizes. Only because fluorescein, especially as carboxyfluorescein diacetate (CFDA) it is widely used, because it crosses membranes, allowing for non-invasive observations.

We agree that the text was insufficiently clear at this point. We have added a few words emphasizing that the size limitation relates to the non-invasive techniques and additionally emphasizing the wider size range of microinjection/bombardment methods.

Modern tracer-based approaches using confocal microscopy can yield very similar results. Compare Rutschow et al., 2011 and Liesche and Schulz, 2013. See also Liesche and Schulz (2012, Plant Physiology), who compared permeabilities across plant species and different cell-cell interfaces.

We assume the Liesche and Schulz, 2013, is the following 2012 manuscript: https://doi.org/10.1111/j.1365-2818.2011.03584.x We have added a reference to this manuscript stating values in the same range (and some lower ones).

Photoactivation and photobleaching approaches are not time consuming. Quite the opposite. Application of tracer, sample preparation and imaging can easily done within half an hour.Introduction paragraph six: I suggest formulating with greater care. Photoactivation and photobleaching approaches as well as GFP transport studies have been very valuable tools for assessing PD function.

We have rephrased the sentence to make it correct: time consuming now only relates to the generation of transgenic lines with cell specific fluorescent probes (as originally intended).

The incompatibility here could have very different reasons. A better citation would be Liesche et al., 2019, which actually tries to compare structure-based modeling with functional data. As mentioned above, the Liesche and Schulz, 2013, paper had a different objective.

Agreed. Reference changed.

General comment to the Introduction: I advise a more nuanced view at the current state of the field and approaches carried out in the past. Also, I recommend using the Liesche et al., 2019 paper in Plant Physiology as starting point because it actually is the first systematic comparison of PD structure and function. Moreover, this publication clearly demonstrates the need for better models.

In response to this general comment, we have made the following changes.

1) Including a more extensive overview of modelling efforts, including manuscripts focussing on flow through PDs. In this, we have classified the models based on the general PD architecture used: an unobstructed sleeve surrounding a desmotubule (the majority) and rarer geometries:

– Funnel shape (only Ross-Elliott et al., 2017) in the phloem unloading zone.

– The sub-nano channel architecture (Liesche and Schulz, 2013 and Comtet et al., 2017), where we also mention the specific context of size selectivity for small molecules.

2) Rephrasing the limitations of modern techniques.

3) Towards the end of the Introduction, we have used Liesche et al., 2019 to illustrate the need for better models, as suggested above.

In processing all reviewer comments, we also realized another issue that hampers the coupling of tissue level and ultrastructural findings: the tissue level measurements are not always reported in (SI) units that can be compared to computed effective symplasmic permeabilities. We have now mentioned this in the Discussion.

Figure 1: The effective symplasmic permeability in a file of cells also depends on cell size and properties of the cytosol.

As we use it, effective symplasmic permeability is a property of the cell wall only. We use cell length L (which is proportional to cell size in a file of rectangular cells) to compute the correction factor f_ih_, but this completely cancels out of the equations. Consequently, P(α) is fully determined by PDs/wall properties (and particle size). Therefore, cell size and cytosol properties are not included in Figure 1. To emphasize the above, we have now explicitly stated this aim when introducing the model setup.

Final paragraph of the Introduction: Python is a start, but how about making a standalone app? Or an add-in/macro for Excel? Something that any molecular biologist (and especially students) can use! Last year I got quite a bit of experimental data on phloem transport and I wanted to check it against theoretical models that have been described as easy to use. My basic Matlab skills were not enough to make it work. In my opinion, if you are serious about getting a lot of scientists to try the model, you have to provide a truly easy-to-use solution.

To increase the ease of use, we have developed a graphical user interface (GUI) that aids the user in creating a correct parameter file (all required numbers etc can be entered in the GUI, which also performs basic verification) and then allows the user to run PDinsight by clicking a button. The GUI is written in TKinter, which is included in any standard Python installation. We have also explored the option of an excel macro/add-in, but we concluded this would be too inflexible and hard to maintain. In the future, we will continue to think about ways to improve the program, working both on easier user interaction and adding/adjusting core functionality to the demands of real data. We already implemented additional features to optimally analyze the two Liesche et al., 2019 derived data sets mentioned above.

Compared to the version uploaded for review, we have made the following improvements:

– GUI for easier access.

– Improved documentation (currently available as appendix to the manuscript).

– Default parameter files for each of the modes, containing only those parameters that actually are used in the specific computation.

– Added a mode for computing a ``unit permeability’’ (\Pi(\α) = “permeation constant of a single PD” = equation 4, in the manuscript), useful for comparisons if density information is lacking.

– Removed the dependency on numpy (if installed, numpy is still used), so the user does not have to install any python modules to make the program work.

Results subsection “Outline of the model”: Many of the equations used in this model were applied to calculate PD transport for the first time by Liesche and Schulz, 2013,. Maybe that should be acknowledged somewhere in this paragraph or the Introduction.

At a conceptual level, this is now acknowledged in the Introduction when mentioning the use of an unobstructed sleeve architecture.

Many equations follow from the geometrical assumptions, but the use of hindrance factors is a point where models differ a lot. So, for this crucial point we added a reference to Liesche and Schulz, 2013 when introducing the use of hindrance factors.

Figure 3: What is the reasoning behind plot B? How can I see from the plot that D always has a big influence (which, of course, is not unexpected)? All values of permeability are very low.

The reasoning behind plot B is to show that the particle’s diffusion constant D has a very large impact on permeability and explains a large part of its size dependence. This is visible in plot B by seeing similar slopes and, for sufficiently large *R_n_*, also values among the curves. For the largest *R_n_*, there is only a 3-fold difference left between 0.1 nm and 2 nm particles (in B), whereas the difference is 50-60-fold in the more realistic plot A. We have added a line in the text describing Figure 3B with this explanation: “For example, at $R_n=R_c$, the 50+ fold difference between $\α=0.1$ nm and $\α=2$ nm is reduced to a 3-fold difference.”

Re low values: All values are in arbitrary units. Therefore, their absolute value is meaningless. We have added a sentence to emphasize this.

Results paragraph five: density jump is not a term often used in PD literature. You could consider using concentration potential.

We have opted for “concentration difference”.

Subsection ““Necked” PDs increase molecular flux in thicker cell walls”: I find this heading slightly misleading. Shouldn't it be "dilated PDs increase molecular flux"? Because R_n_ is the same in both cases, meaning that the neck doesn't change. Or am I just misled by the drawing in Figure 4?

Agreed, changed to “A dilated central region increases molecular flux in thicker cell walls”.

First paragraph of subsection ““Necked” PDs increase molecular flux in thicker cell walls”: Maybe write "electron microscopy" instead of "evidence".

Agreed, changed as suggested.

Figure 4: Again, "dilated" seems more appropriate than "necked". Panel C has a different y axis scale from all the other panels. The legend in panel C is confusing as it is missing an α. I recommend either legends in all panels or description in the figure caption.Figure 4 and accompanying text: Is R_n_ = 12nm in the 'necked' PD and 17.5 in case of narrow PD? This should be formulated clearly, both in the figure and the text. The drawing above panel A gives the impression that they have the same R_n_.

The neck diameter in all cases is 12 nm as the drawing on top suggests. We have clarified this in the text and by adding a sentence to the beginning of the caption of Figure 4.

We have changed the top to “dilated” and added α to the legend (also in Figure 4—figure supplement 1). We have not changed the y axes of all panels, because that would reduce visibility. This change also affects the parameter Q_rel_ and \tau_rel_.

Paragraph three of the same section: In this case the length of the neck region is still 1/3rd of the total PD length, right? It might be worthwhile to also check the effect of variation in the length of the necked region. Ideally, this should also be one parameter of the python tool, because it can be estimated from TEM images.

Added a figure (Figure 4—figure supplement 2) exploring the effects of neck region length. Neck length can already be entered as a parameter in PDinsight.

Subsection “The desmotubule increases PD transport and changes the dependence on particle size”: I also find this heading misleading. My first intuition would be to compare a PD with DT to a PD without DT.

Changed heading, emphasizing that PDs with a given maximum particle size (~SEL) are compared. Note that we briefly investigate the effect of DT removal in Figure 8CD in the context of the Rutschow et al., 2011 experiments.

In the same subsection what exactly does the assumption on selection towards size-selectivity mean? That a plant would make the PD as narrow as possible, depending on the molecule that is supposed to pass? This assumption might be valid for relatively big molecules, but probably not for small ones (below 1-2nm). Except for the very special PDs in the polymer trapping species mentioned earlier. I recommend discussing this assumption in more detail.

Size selectivity is meant in the context of the model: control over the maximum particle size that can pass. This could yield a requirement that at a certain developmental stage, only particles of, say, <2.5 nm radius should be allowed to pass, which in our model translates directly to a 5 nm difference between Rdt and *R_n_*, or, for cylindrical channels without DT, a radius of 2.5 nm.

To avoid confusion (the same point was also raised by reviewer 2), we have rephrased the sentence and removed the reference to natural selection.

Where exactly does the "see methods" refer to? I don't find a section on DT-related calculations in the methods.

Reference removed. The full calculation is included in the main text.

Paragraph two of subsection “Clustering of PDs in pit 1elds reduces effective symplastic permeability”: Wasn't R_n_ = 12 nm in Figure 4?

Corrected.

In paragraph three of the same section: Rpit is not in the list of mathematical symbols. Moreover, it is not clear why the radius of a pit field should be comparable to the radius of individual PD, since the number and size of PD within a pit field can vary.

Added Rpit to the list. Also added a few words to emphasise that Rpit is much larger than *R_n_* (except for the trivial case with a single PD in a “pit field” that no biologist would call pit field).

In the final paragraph of the same section: How should this be measurable? As a correlation of clustering with wall thickness?

The aim of the sentence was to emphasize that for realistic *R_n_*, the effects of Figure 6B,D would be too small to be experimentally measurable in any way, followed by the observation that PD length and density have more impact on fih when PDs are clustered into pit fields. We have rephrased the sentence to avoid this confusion.

We agree that measuring fih experimentally is currently impossible, as the discrepancies in permeability values from bleaching/photoactivation experiments and calculated from EM observations are too large with current techniques. This may remain so for a long time, and even if technically possible, the question is who would be interested in measuring this correction factor independently.

Figure 7: In panel A, is the distance between PD or the radius of the pit field constant?

Distance between PDs. Added a few words to clarify. The cartoons are indeed suggesting otherwise, which is hard to prevent without using cartoons that are too large.

Paragraph three of subsection “Application of the model to compute effective permeability for fluorescein derivatives”: With a density of 5 (as specified by Zhu et al., see comments below), it seems like the value for R_n_ should be 22 nm.

Required *R_n_* would be 16.5, 16.9, or 18.3 nm, depending on the assumed PD density (5.42/ 5 / 4 PD μm^2^, respectively), a target value of P=3.3 um/s and otherwise the same parameters as in our main text. We have now also included the calculations for this second experiment in the main text and Table 1. Initially, we had not included this second experiment in our manuscript, as for geometrical reasons the interpretation of the single cell bleach is more complicated than the tissue level bleach. This is illustrated by the fact that Rutschow et al. chose to neglect any coupling in the radial and tangential directions (detailed in the supplementary text of Rutschow et al).

Paragraph four: Why is a density of 10 μm^2^ used here instead of the value from Zhu et al. (5.4 μm^2^)?

Zhu et al. Table 2 reports 5.4 density values for the epidermis whereas the reported values for tangential walls of inner cortex (12.58), outer cortex (9.05) and (and similar to vascular (9.92)) were in the range of densities used here (10 μm^2^ -13 μm^2^). The quoted targets of P=6-8.5 um/s were based on the tissue level bleaches, which involve multiple cell layers and hence include only a limited fraction of epidermal cells (Figure 1 Rutschow et al).

Rutschow et al. also report measurements for single epidermal cells, where the value of 5.42 would be applicable, possibly weighted with the radial value of 3.33 μm^2^. For this measurement (Figure 3 Rutschow et al), they report P=3.3 +/- 0.8 um/s. (also see previous comment).

To clarify this issue we have explicitly included the exact measurements of Zhu et al.. We have also included calculations for the epidermal experiment, there using a density of 5 (or 5.42) μm^2^. The predictions are remarkably consistent.

Also didn't a previous paragraph show that central dilation ("necking") is especially effective for long PDs? How does that fit with the results presented here?

Unfortunate phrasing, rephrased. The point is that central dilation can increase permeability (most effectively in long PDs), but only to a limited extent (a straight channel with radius Rc). The Rutschow values cannot be reproduced by central dilation.

Table 1: PD density seems to start at 10 μm2. This value is higher than the 5.4 μm2 specified by Zhu et al. (Table 2). Their value should be considered a max value as the area that was analyzed by Rutschow et al. is reaching into the mature root zone already. Mature zone has much lower density values (0.62 μm2, Table 3 in Zhu et al.).

See above: the value of 5.4 μm^2^ is not applicable to the experiment we analyzed. As we have now also analyzed the epidermal experiment (with a target effective symplasmic permeability of 3.3 um/s), we have also added 5 and 5.42 μm^2^ to Table 1B.

The Figure 1 experiment in Rutschow et al. was performed in a 50 μm zone at ~200 μm from the QC. Cell shapes seem not elongated yet (eg, see Figure 3 in the Rutschow paper), so likely before the “transition zone” (after Sabatini).

The measurements for immature tissue in Zhu et al. were performed at 100 / 150 μm from the QC, depending on cell type / interface (not 100% clear from the manuscript which was used for each measurement). Their measurements for mature tissue were performed at 1200 – 1250 μm from the QC.

Taken together, while we appreciate the concern, it seems likely that the densities in the Rutschow experiment are closer to the immature than the “mature” densities in Zhu et al. We have, therefore, not added the low “mature” densities to Table 1.

Another way to look at the data is that the root zone used in the Rutschow experiment has to have “immature” PD densities if PD transport is predominantly diffusive in this zone/direction, because otherwise absurd PD densities and/or *R_n_* would be required for reproducing the measured permeabilities assuming only diffusive transport.

In paragraph five the model seems incompatible with the low-concentration H_2_O_2_ data. At a density of 5.4 μm^2^, R_n_ would need to increase to over 40 nm according to Figure 8B. This seems unrealistic. An increase in PD number can also not be expected because of the short treatment period of 2h.

See comments above: a density of ~10 μm^2^ is more likely and the speed of the sink-source transition in (tobacco) leaves and corresponding changes in PD structure indicate that a density increase could occur in a short window of time, although such a massive increase in 2 hours has not been reported under natural circumstances.

That said, explaining these data seem to require a dramatic change at the PD level of whatever kind. We have altered the text and more explicitly in the Discussion, to emphasize this point.

“We also compared the results obtained with our model and the sub-nano channel model reported before (Liesche and Schulz, 2013).”: This makes very little sense as this model was put forward only for a very specific case, as mentioned above. It would be better to compare with the cytosolic slit model that was also described in that paper.

We agree that comparison to other cytosolic sleeve models is important, so we have added this in the Discussion. The comparison with the sub-nano channel architecture is valuable because it is also used by others (Comtet et al., 2017), and as a (pessimistic) lower bound for the performance of a sleeve with spokes. We have rephrased the text and removed the reference to Liesche and Schulz at this point to emphasize that the comparison here is about different possible geometries, not specific instances of the model. Also see the next comment.

Discussion paragraph two: Again, this model should not be compared here or it should be clearly stated that it was not developed to be applied in this way. Instead the authors should compare their model to the cytosolic sleeve model of Liesche and Schulz, 2013, or the 'pure diffusion' model by Doelger et al., 2014 or the model for 'diffusion through simple PD' included in Ross-Elliott et al., 2017.

We have kept a short comparison between sub-nano channel and unobstructed sleeve _architecture_, which we subsequently use to state that the sub-nano calculations provide a (pessimistic) estimate for the effect of spokes in the cytosolic sleeve. To avoid the suggestion that the sub-nano channel model was developed to explain the Rutschow et al. data, we compare “architectures” and do not reference Liesche et al., 2013 here.

In the two paragraphs above, we have included a comparison with other models: the three straight unobstructed sleeve models mentioned by the reviewer 629-632) and the central dilation geometry used by Blake, 1978 (who only studies flow, not diffusion).

Differences in model PD geometry are also detailed in the Introduction.

“For example, sucrose moves symplastically from bundle sheet cells (BSC) to intermediary cells (IC), where it is polymerized into the larger oligomers raffinose and stachyose, that do not diffuse back in detectable amounts”: It should be mentioned that this only happens in certain species (active symplasmic phloem loaders) with the Cucurbits as the most prominent example.

Mention added.

Discussion paragraph four: The Liesche et al., 2019, paper does not analyze correlation of PD length with permeability at the BSC-phloem interface of active symplasmic loaders. It analyzes correlations of PD anatomical parameters (including length) with permeability across species. I don't understand why a correlation of PD length with permeability would be expected, as the study generally finds permeability to depend to a much higher degree on PD diameter. Please also note that it is very unlikely that a bulk flow was overlooked in that study as the measurements were performed on detached leaves. Indeed, this has been tested by Liesche and Schulz, 2012 (Plant Physiology) who compared full photosynthesising leaf and cut-out tissues and found no difference in PD permeability in tobacco.

Considering the issues mentioned above as well as other issues that came up during subsequent discussion among us authors, we have decided to remove the sentence and reference here.

Discussion paragraph six: please include 'at the root unloading zone'. Funnel PDs are not found at other phloem interfaces.

Added.

“Applying our model for diffusion as a sole driver of symplastic transport can indeed explain experimentally observed measurements of effective symplastic permeability for CF, but only with somewhat wider PDs/neck regions or several fold higher PD densities than usually measured by EM.”: This statement seems to contradict itself. In addition, I highly recommend application of the model to additional interfaces. Liesche et al., 2019, for example, found a model that matches observations of permeability for the bundle-sheath-phloem interface, but not for the bundle sheath-mesophyll interface. In the present case, it should be noted that PD density and dimensions were determined for sand-grown plants (Zhu et al., 1998), whereas effective diffusivity was measured for plate-grown plants (Rutschow et al., 2011). I don't know how growth conditions affect PDs, but it has been previously shown that there are big differences, for example in gene expression, between soil- and plate-grown plants. I mention this just to illustrate the need for additional tests.

We agree that testing more interfaces is important. We have added a section emphasizing the need for and benefits of additional testing following the discussion of this experiment. Moreover, we have modified the discussion structure, in the process of merging both points where the Rutschow et al., 2011 experiments are discussed. As mentioned, we now have analyzed both Rutschow experiments, one involving an entire root section, the other only the epidermis. Additionally, we have analyzed data from two mesophyll-mesophyll interfaces in poplar and Cucurbita max that were originally acquired in the context of Liesche et al., 2019, but not published in the detailed form that we needed. Results of this analysis are included in a separate document for the reviewers/editors, but are not meant to be included in the current manuscript to avoid a possible conflict of interest.

“Our model can also explain the effect on permeability after treatment with high and low concentrations of H_2_O_2_ in Rutschow et al., 2011.”: Please reconsider this statement regarding low H_2_O_2_ in light of comments above.

We have rephrased the statement such that high H_2_O_2_ can easily be explained, whereas low H_2_O_2_ is much harder. We have also rewritten the remainder of the section and omitted the references mentioned below.

I am very skeptical about the conclusion. Stress perception, signaling, and two rounds of twinning, all within 2h seem unrealistic. I also didn't find evidence for such a rapid multiplication of PD numbers in the two citations provided here. Moreover, 30 PD μm^2^ would be an extremely high number, that, to my knowledge, has never been observed at a "regular" cell-cell interface.

We agree that the 30+ density is indeed very high. Then again, regular cell-cell interfaces are not regularly exposed to 2 hours of H_2_O_2_ signaling. The only way to find out is to prepare similarly treated tissue for EM investigation. If the density is indeed dramatically increased, this should be relatively easy to see. We have added this suggested experiment to the text.

Additionally, we have added some more skepticism in the discussion of the low H_2_O_2_ results. The conclusion of (limited) evidence in those papers is based on the following observations: Roberts et al., 2001: formation of complex PDs coincides with the leaf sink-source transition in tobacco leaves. Particularly the walls of pallisade and spongy parenchyma show sharp transitions in the occurrence of branched PDs (Figure 5 of Roberts et al.). At a macroscopic level, the sink-source transition progressed over the leave with speeds of 100 um/h (small leaves, avg 4.5 cm), 310 um/h (medium leaves, avg 7 cm), 600 um/h (large leaves, avg 10 cm) (measured using AtSuc2-GFP), i.e., in all cases involving many cells per hour. → With a speed of multiple cell walls per hour, PD architecture might be changed from simple to predominantly complex, which involves a large increase in the number of PD entrances at the transformed PDs.

Fitzgibbon et al., 2013: “With young detached leaves (leaves 1 and 2), we noticed that the conversion of simple to complex PDs occurred rapidly. New complex PDs could be visualized within 1 h of leaf detachment and were formed continuously at an average rate of 0.18 PDs per cell per hour (Figure 2G). In the areas observed, the overall numbers of PDs increased two- to threefold within 48 h.” → formation of complex PDs (from simple), which may have more than 2 channels, is possible within 1 h. The H_2_O_2_ treatment would have to synchronize the process and induce it in parallel at many PDs in the same interface, as the “natural” rate of 0.18 PDs per cell per hour is far insufficient.

We thought that this consideration was too detailed and too long for something that produces a mere hint that the required density might be possible (but certainly no strong evidence) to be included in the text. Therefore, we have deleted the references to the above papers.

Discussion paragraph seven, “Despite these deviations, comparing our model to the sub-nano channel model, we found that the latter requires roughly twice as high PD densities to produce the same permeability values P (CF)” and following sentences: Again, this is not an appropriate comparison. Please compare to other models listed in a comment above.

We have restructured the Discussion, in the process reducing emphasis on the difference between sub-nano and unobstructed sleeve architecture, as well as including more comparison with other unobstructed sleeve models including the ones above. As mentioned before, we think that the comparison with the sub-nano channel architecture is valuable, so not removed.